# Complexity Lower Bounds for Nonconvex-Strongly-Concave Min-Max Optimization

**Haochuan Li**
Department of EECS
MIT
Cambridge, MA 02139
haochuan@mit.edu

**Yi Tian**
Department of EECS
MIT
Cambridge, MA 02139
yitian@mit.edu

**Jingzhao Zhang**
Department of EECS
MIT
Cambridge, MA 02139
jzhzhang@mit.edu

**Ali Jadbabaie**
Department of CEE
MIT
Cambridge, MA 02139
jadbabai@mit.edu

## Abstract

We provide a first-order oracle complexity lower bound for finding stationary points of min-max optimization problems where the objective function is smooth, nonconvex in the minimization variable, and strongly concave in the maximization variable. We establish a lower bound of $\Omega\left(\sqrt{\kappa}\epsilon^{-2}\right)$ for deterministic oracles, where $\epsilon$ defines the level of approximate stationarity and $\kappa$ is the condition number. Our lower bound matches the best existing upper bound in the $\epsilon$ and $\kappa$ dependence up to logarithmic factors. For stochastic oracles, we provide a lower bound of $\Omega\left(\sqrt{\kappa}\epsilon^{-2} + \kappa^{1/3}\epsilon^{-4}\right)$. It suggests that there is a gap between the best existing upper bound $\mathcal{O}(\kappa^3\epsilon^{-4})$ and our lower bound in the condition number dependence.

## 1 Introduction

In this paper, we study the oracle complexity lower bound of the following min-max optimization problem:

$$\min_{\boldsymbol{x}\in\mathcal{X}} \max_{\boldsymbol{y}\in\mathcal{Y}} f(\boldsymbol{x};\boldsymbol{y}), \tag{1}$$

where $\mathcal{X} \subset \mathbb{R}^m$ and $\mathcal{Y} \subset \mathbb{R}^n$ are nonempty closed convex sets. Such a problem arises in a wide range of applications, e.g., two-player zero-sum games [38], Generative Adversarial Networks [17], robust optimization [4], including defending adversarial attacks [18, 29].

The research in min-max problems (1) has a long history. If $f$ is linear in both $\boldsymbol{x}$ and $\boldsymbol{y}$, the problem is known as bilinear min-max optimization, for which von Neumann's min-max theorem [38] guarantees the existence of a saddle point, a point $(\boldsymbol{x}^*, \boldsymbol{y}^*)$ that satisfies strong duality (i.e. min-max equals max-min). Later, Sion [44] generalized the statement to convex-concave functions under certain regularity conditions, which led to the theory of KKT conditions. Aside from studying the existence of strong duality, Nemirovski [33], Nesterov [36] proposed algorithms that can find approximate saddle points for convex-concave objectives with an $\mathcal{O}(1/\epsilon)$ convergence rate. The rate was proven to be worst-case optimal even for min-max problems with a bilinear cross-term [41].

Going beyond the assumption of convex-concavity poses a big challenge. In general nonconvex-nonconcave min-max optimization problems, a saddle-point may not exist [22]. Defining a notion of min-max points that is simultaneously nontrivial and tractable is still an open question. If one uses the

35th Conference on Neural Information Processing Systems (NeurIPS 2021).

standard definition of saddle points as in the strong duality problems, then determining its existence is known to be NP-hard [9], and finding an approximate local saddle point is PPAD-complete [9]. Alternatively, a relaxed definition of min-max point known as Stackelberg equilibrium is guaranteed to exist [22], yet only local asymptotic convergence is known [22, 47].

Due to the lack of a well-formulated suboptimality measure for general nonconvex-nonconcave problems, recent research has considered problems with special structures. For example, Lin et al. [26, 27] studied nonconvex-concave problems, Yang et al. [50] studied problems satisfying the Polyak-Lojasiewicz inequality, Diakonikolas et al. [11] studied problems with weak Minty variational inequality solutions, Mangoubi and Vishnoi [30] considered an algorithm specific equilibrium.

Our work focuses on the analysis of gradient oracle complexity in the nonconvex-strongly-concave min-max setting. In the standard convex-concave setup, Lin et al. [27] designed a set of algorithms that achieve near-optimal gradient complexity for all the three variants of convex-concave min-max optimization. However, in the nonconvex-concave setting, it is unclear whether their algorithm is optimal given that no matching lower bound exists. In this work, our main contribution is as follows:

- We construct an explicit example on which no first-order zero-respecting algorithm can achieve less than $\Omega(\sqrt{\kappa}\epsilon^{-2})$ oracle complexity for smooth nonconvex-strongly-concave problems, matching the $\tilde{\mathcal{O}}(\sqrt{\kappa}\epsilon^{-2})$ complexity in [27].

- We further extend our results to the stochastic(-oracle) setting, achieving the complexity lower bound $\Omega(\sqrt{\kappa}\epsilon^{-2} + \kappa^{1/3}\epsilon^{-4})$. Compared against the results in [26], our result suggests that the dependency on the condition number may not be optimal.

## 1.1 Related work

**Upper bounds.** Since standard gradient descent-ascent diverges for even the simplest case of bilinear zero-sum games, researchers proposed different variations to achieve convergence in the convex-concave setting, with possibly additional structures. One of the variations is known as extra-gradient method [24, 46, 33, 7, 49], which takes two gradient steps per iteration. Another line studies the optimistic gradient method [43, 8, 15, 31, 20]. Mokhtari et al. [32] proposed a unified analysis through the lens of proximal update. Some works [1, 27] focused on accelerating the known rates in terms of the conditional number dependence. Golowich et al. [16] studied last iterate convergence. Lee et al. [25], Daskalakis et al. [10], Wei et al. [48] studied two-player zero-sum stochastic games.

Convergence under the nonconvex-concave setup is less studied. Rafique et al. [42] studied a proximal version of gradient methods. Lin et al. [26] studied gradient descent ascent with different time scales. Thekumparampil et al. [45] proposed an implicit algorithm for the nonconvex-concave setting. Many other works also studied different variations of convergence [39, 23, 28, 40].

**Lower bounds.** Our work follows the complexity framework introduced by Nemirovski in the 1980s. The most classical lower bounds are due to Nemirovski and Nesterov, and are discussed in their textbooks [35, 37]. Recently, much progress was made [13, 5, 6, 3] by extending the original analysis to the nonconvex setting. Our work builds upon these results. Another line of works studies lower bounds for quadratic problems in min-max setting and utilizes a different framework sometimes known as Stationary Canonical Linear Iterative (SCLI) [2, 41, 51, 21]. It exploits the closed-form updates for linear dynamics and studies convergence properties via the transition matrices.

Concurrently, Zhang et al. [52] and Han et al. [19] provided lower complexity bounds similar to ours in the deterministic nonconvex-strongly-concave setting. Their constructions are similar to each other but different from ours. Their constructions can be roughly viewed as a simple addition of the quadratic instance in [41] and a nonconvex component as in [6]. Informally speaking, their $\sqrt{\kappa}$ factor comes from the rate in [41]. However, our construction is mainly inspired by the chain-like instance in [5]. To obtain the $\sqrt{\kappa}$ factor, we insert quadratic sub-chains into it to increase its length by a factor of $\sqrt{\kappa}$. One possible advantage of our construction is that the minimization and maximization variables are weakly coupled so that we may have more freedom to modify the component functions to extend our results to other settings such as the nonconvex-concave function class. Zhang et al. [52] and Han et al. [19] also provided novel analyses with extensions to the upper and lower bounds in the finite-sum setting, whereas we further extend our results to the stochastic setting.

## 2 Preliminaries

Before introducing our main results, we describe the problem setup and algorithm complexity in this section. First, we introduce the notation. Next, we define the function class and algorithm class. Finally, we provide the formal definition of optimality measure and gradient complexity.

**Notation.** We use bold lower-case letters to denote vectors and use $x_i$ to denote the $i$-th coordinate of vector $\boldsymbol{x}$. Let $\operatorname{supp}(\boldsymbol{x}) := \{i : x_i \neq 0\}$ denote the support of $\boldsymbol{x}$. Let $\|\boldsymbol{x}\|_2$ and $\|\boldsymbol{x}\|_\infty$ denote its $\ell_2$ and $\ell_\infty$ norm respectively. For a matrix $A$, we use $A_{i,j}$ to denote its $(i,j)$-th entry and $A^\top$ its transpose. We use $\|A\|_2$ to denote its spectral norm and $\det(A)$ its determinant. We use calligraphic upper-case letters to denote sets, as in $\mathcal{X}$ and $\mathcal{Y}$. For a nonempty closed convex set $\mathcal{X} \in \mathbb{R}^d$, let $\mathsf{P}_\mathcal{X}$ denote the Euclidean projection onto $\mathcal{X}$. We also use a semi-comma besides commas to split minimization and maximization variables of a function. For example, when we write $f(\boldsymbol{x}, \boldsymbol{z}; \boldsymbol{y})$, it means $\boldsymbol{x}$ and $\boldsymbol{z}$ are the variables to minimize and $\boldsymbol{y}$ is the variable to maximize. Finally, we use the standard $\mathcal{O}(\cdot)$ and $\Omega(\cdot)$ notation, with $\tilde{\mathcal{O}}(\cdot)$ and $\tilde{\Omega}(\cdot)$ further hiding log factors.

### 2.1 Function class

Our analysis focuses on the class of smooth nonconvex-strongly-concave functions defined below.

**Definition 1.** *Given $L \geq \mu > 0$ and $\Delta > 0$, we use $\mathcal{F}(L, \mu, \Delta)$ to denote the set of all functions $f : \mathcal{X} \times \mathcal{Y} \to \mathbb{R}$ for some nonempty closed convex sets $\mathcal{X} \subset \mathbb{R}^m$ and $\mathcal{Y} \subset \mathbb{R}^n$ where $m, n \in \mathbb{N}$, which satisfies the following assumptions:*

1. *$f$ is $L$-smooth, that is, for every $\boldsymbol{x}_1, \boldsymbol{x}_2 \in \mathcal{X}$ and $\boldsymbol{y}_1, \boldsymbol{y}_2 \in \mathcal{Y}$,*
$$\|\nabla f(\boldsymbol{x}_1; \boldsymbol{y}_1) - \nabla f(\boldsymbol{x}_2; \boldsymbol{y}_2)\|_2 \leq L \|(\boldsymbol{x}_1, \boldsymbol{y}_1) - (\boldsymbol{x}_2, \boldsymbol{y}_2)\|_2 \, ;$$

2. *For any fixed $\boldsymbol{x} \in \mathcal{X}$, $f$ is $\mu$-strongly concave in $\boldsymbol{y}$, that is, for any $\boldsymbol{y}_1, \boldsymbol{y}_2 \in \mathcal{Y}$,*
$$f(\boldsymbol{x}; \boldsymbol{y}_1) \leq f(\boldsymbol{x}; \boldsymbol{y}_2) + \nabla_{\boldsymbol{y}} f(\boldsymbol{x}; \boldsymbol{y}_2) \cdot (\boldsymbol{y}_1 - \boldsymbol{y}_2) - \frac{1}{2} \mu \|\boldsymbol{y}_1 - \boldsymbol{y}_2\|_2^2 \, ;$$

3. *$f_m(\boldsymbol{0}) - \min_{\boldsymbol{x} \in \mathcal{X}} f_m(\boldsymbol{x}) \leq \Delta$, where $f_m(\boldsymbol{x}) := \max_{\boldsymbol{y} \in \mathcal{Y}} f(\boldsymbol{x}; \boldsymbol{y})$.*

For any fixed $\boldsymbol{y} \in \mathcal{Y}$, $f$ is potentially nonconvex in $\boldsymbol{x}$. Note that $\mathcal{F}(L, \mu, \Delta)$ includes functions with domain on $\mathbb{R}^m \times \mathbb{R}^n$ for all $m, n \in \mathbb{N}$, following the framework of dimension-free convergence guarantees [34, 37]. In Section 5, we will construct a hard instance with domain dimensions $m, n$ both growing inversely in the required accuracy $\epsilon$.

### 2.2 Algorithm class

In this section, we describe the algorithms of interest for the min-max problems with the function class defined above. In particular, we restrict our analysis to first-order algorithms that optimize objectives using first-order oracles defined below.

**Definition 2** (Deterministic first-order oracle). *The deterministic first-order oracle of a differentiable function $f : \mathcal{X} \to \mathbb{R}$ is a mapping $O : \boldsymbol{x} \mapsto (f(\boldsymbol{x}), \nabla f(\boldsymbol{x}))$ for $\boldsymbol{x} \in \mathcal{X}$.*

**Definition 3** (Stochastic first-order oracle). *A stochastic first-order oracle with bounded variance $\sigma^2$ of a differentiable function $f : \mathcal{X} \to \mathbb{R}$ is a mapping $O : \boldsymbol{x} \mapsto (f(\boldsymbol{x}), \boldsymbol{g}(\boldsymbol{x}, \xi))$ for $\boldsymbol{x} \in \mathcal{X}$, where $\xi$ is a random variable satisfying $\mathbb{E}[\boldsymbol{g}(\boldsymbol{x}, \xi)] = \nabla f(\boldsymbol{x})$ and $\mathbb{E}\left[\|\boldsymbol{g}(\boldsymbol{x}, \xi) - \nabla f(\boldsymbol{x})\|_2^2\right] \leq \sigma^2$.*

We say $O$ is a first-order oracle if it is a deterministic or stochastic first-order oracle.

Furthermore, we consider first-order algorithms satisfying the zero-respecting assumption. Formally, we define first-order algorithms as follows.

**Definition 4** (First-order algorithm). *A first-order (zero-respecting) algorithm is one that for any function $f : \mathcal{X} \to \mathbb{R}$ and its associated first-order oracle $O_f : \boldsymbol{x} \mapsto (f(\boldsymbol{x}), \boldsymbol{g})$, the $(t+1)$-th iterate $\boldsymbol{x}^{t+1}$ satisfies*
$$\boldsymbol{x}^{t+1} \in \left\{ \mathsf{P}_\mathcal{X}(\boldsymbol{v}) : \operatorname{supp}(\boldsymbol{v}) \subset \bigcup_{0 \leq i \leq t} \left(\operatorname{supp}(\boldsymbol{x}^i) \cup \operatorname{supp}(\boldsymbol{g}^i)\right) \right\} .$$

Definition 4 extends the standard zero-respecting algorithms [5, 3] to the constrained setting. It covers most existing first-order methods used in the literature including (projected) stochastic gradient descent, adaptive methods, and more importantly, the algorithms used in [26, 27] which achieve the upper bounds for nonconvex-strongly-concave min-max optimization in the deterministic and stochastic settings, respectively. There is a standard reduction from a lower bound for zero-respecting algorithms to that for arbitrary deterministic algorithms with deterministic or even stochastic first-order oracles [5, 3]. We defer this extension to future work.

### 2.3 Optimality via approximate stationarity

We measure the progress of solving the nonconvex-strongly-concave problem via the gradient norm of the maximized function with respect to the minimization variable $x$. Following [14], we define the notion of $\epsilon$-stationary points in presence of constraints as follows.

**Definition 5.** *Let $\mathcal{X}$ be a nonempty closed convex set. A point $x \in \mathcal{X}$ is said to be an $\epsilon$-stationary point of an $L$-smooth function $f : \mathcal{X} \to \mathbb{R}$ if*

$$L \left\| \mathsf{P}_{\mathcal{X}} \left[ x - (1/L)\nabla f(x) \right] - x \right\|_2 \leq \epsilon.$$

Note that the algorithm studied in [26] also assumes bounded domain and used the same notion of stationarity. This definition of stationary points reduces to $\|\nabla f(x)\|_2 \leq \epsilon$ when $\mathcal{X} = \mathbb{R}^d$ for some $d \in \mathbb{N}$. For any $f \in \mathcal{F}(L, \mu, \Delta)$, Definition 5 applies to $f_m$ as it is $L_m$-smooth, where $L_m \leq (\kappa + 1)L$ by [26, Lemma 4.3].

Our goal of solving the nonconvex-strongly-concave min-max optimization problem is to find an $\epsilon$-stationary point of $f_m$. We show that no deterministic first-order algorithm can achieve less than $\Omega(\sqrt{\kappa}\epsilon^{-2})$ gradient complexity for smooth nonconvex-strongly-concave problems, matching the $\tilde{\mathcal{O}}(\sqrt{\kappa}\epsilon^{-2})$ complexity in [27]. We further extend our result to the stochastic setting, achieving a complexity lower bound $\Omega(\sqrt{\kappa}\epsilon^{-2} + \kappa^{1/3}\epsilon^{-4})$.

## 3 Main results

In this section, we present our main results on the minimum number of gradient oracle calls required to find an $\epsilon$-stationary point. The results for deterministic and stochastic settings are presented in the following two subsections respectively.

### 3.1 Lower bound on first-order oracle complexity in the deterministic setting

Nonconvex-strongly-concave min-max optimization subsumes nonconvex optimization, the lower bound $\Omega(L\Delta/\epsilon^2)$ in nonconvex optimization [5] also holds for nonconvex-strongly-concave min-max optimization. However, compared with the $\tilde{\mathcal{O}}(L\Delta\sqrt{\kappa}/\epsilon^2)$ upper bound [27], a $\sqrt{\kappa}$ factor is missing. Our main result below fills this gap, showing that the known rate by [26] is optimal up to log factors.

**Theorem 1.** *For any $\mu, L, \Delta, \epsilon > 0$ such that $\kappa = L/\mu \geq 1$, there exists a function instance $f : \mathbb{R}^m \times \mathbb{R}^n \to \mathbb{R}$ in $\mathcal{F}(L, \mu, \Delta)$ for some $m, n \in \mathbb{N}$ with its deterministic first-order oracle such that for any first-order algorithm, we have $\|\nabla f_m(x^t)\|_2 > \epsilon$, where $f_m(x) := \max_y f(x, y)$, whenever*

$$t \leq \frac{c_0 L\Delta\sqrt{\kappa}}{\epsilon^2},$$

*where $c_0$ is a numerical constant.*

Similar to the lower bound in [5], our lower bound applies to dimension-free optimization in nature. That is, for a given $\epsilon$, we construct a hard instance with dimension $d = \mathcal{O}(1/\epsilon^2)$, which can be very large if $\epsilon$ is small. The discussion on the proof is deferred until Section 5.

### 3.2 Lower bound in the stochastic setting

In the stochastic setting, Arjevani et al. [3] provided a lower bound of $\Omega\left(L\Delta\sigma^2/\epsilon^4\right)$ for smooth nonconvex optimization which is a special case of nonconvex-strongly-concave min-max optimization. Our analysis improves this bound by a factor of $\kappa^{1/3}$ in Theorem 2.

**Theorem 2.** *For any $\mu, L, \Delta, \epsilon, \sigma > 0$ such that $\kappa = L/\mu \geq 1$, there exists a function instance $f : \mathcal{X} \times \mathcal{Y} \to \mathbb{R}$ in $\mathcal{F}(L, \mu, \Delta)$ and a stochastic first-order oracle $O$ for $f$ with variance $\sigma^2$ such that for any first-order algorithm, we have $\mathbb{E}\left[L_m \left\| \mathsf{P}_{\mathcal{X}} \left[ \boldsymbol{x}^t - (1/L_m)\nabla f_m(\boldsymbol{x}^t) \right] - \boldsymbol{x}^t \right\|_2 \right] > \epsilon$, where $f_m(\boldsymbol{x}) := \max_{\boldsymbol{y}} f(\boldsymbol{x}, \boldsymbol{y})$, whenever*

$$t \leq c_0 L \Delta \left( \frac{\sqrt{\kappa}}{\epsilon^2} + \frac{\kappa^{1/3}\sigma^2}{\epsilon^4} \right)$$

*where $L_m$ is the smoothness parameter of $f_m$ and $c_0$ is a numerical constant.*

Lin et al. [26] provided an upper bound of $\mathcal{O}\left(\kappa^3 \epsilon^{-4}\right)$. Therefore, there is a gap between our lower bound and their upper bound in terms of the dependency on $\kappa$.

In summary, our proposed lower bounds improve the known ones by a multiplicative dependence on the condition number. Before proceeding to discuss the concrete techniques, we first summarize the general framework for establishing lower bounds in the next section.

# 4 Framework for proving lower bound

In this section, we provide an outline for the proof of the lower bound in [37] and [5], which lay the foundation for our construction of the hard instance. Both works utilize the notion of zero-chains [5], which instantiates a class of hard functions for optimization. We first define a zero-chain and then discuss how it is used to establish complexity lower bounds.

**Definition 6** (Zero-chain). *We say a function $f : \mathcal{X} \to \mathbb{R}$, where $\mathcal{X} \subset \mathbb{R}^d$, is a (first-order) zero-chain if for every $1 \leq i \leq d$,*

$$\operatorname{supp}(\boldsymbol{x}) := \{i : x_i \neq 0\} \subset \{1, \ldots, i-1\} \implies \operatorname{supp}(\nabla f(\boldsymbol{x})) \subset \{1, \ldots, i\}.$$

Suppose the domain $\mathcal{X} \subset \mathbb{R}^d$ satisfies $\operatorname{supp}(\mathsf{P}_{\mathcal{X}}(\boldsymbol{x})) = \operatorname{supp}(\boldsymbol{x})$ for all $\boldsymbol{x} \in \mathbb{R}^d$, i.e., projecting $\boldsymbol{x}$ onto $\mathcal{X}$ does not change its support. For example, this requirement holds when $\mathcal{X}$ is a hypercube or the whole space $\mathbb{R}^d$. If we run a first-order algorithm on a zero-chain initialized at $\boldsymbol{x}^0 = \mathbf{0}$ (which we assume to hold without loss of generality) with a deterministic first-order oracle, then at each iteration, at most one new coordinate of $\boldsymbol{x}$ becomes nonzero ("discovered"). Therefore, $\operatorname{supp}(\boldsymbol{x}^t) \subset \{1, \ldots, t\}$. Then we obtain a lower complexity bound of $T$ supposing we can show a good solution exists only if at least $T$ coordinates are discovered.

Therefore, the key to proving a lower bound of first-order algorithms with a deterministic oracle is to find a function $f$ such that:

1. it is a zero-chain that belongs to the function class we are interested in; and that

2. we cannot obtain an $\epsilon$-optimal solution if the $t$-th coordinate of $\boldsymbol{x}$ is zero for every $t \geq T$.

This is actually a general strategy for proving lower bounds of first-order methods, used in the lower bound construction both by Carmon et al. [5] for smooth nonconvex optimization and by us here for nonconvex-strongly-concave min-max optimization.

In the stochastic setting, we utilize the generalized notion known as the probability-$p$ zero-chain [3] to prove a lower bound.

**Definition 7** (Probability-$p$ zero-chain). *A function $f : \mathcal{X} \to \mathbb{R}$ with a stochastic first-order oracle $O : \boldsymbol{x} \mapsto (f(\boldsymbol{x}), \boldsymbol{g}(\boldsymbol{x}, \xi))$ is a probability-$p$ zero-chain if*

$$\operatorname{supp}(\boldsymbol{x}) \subset \{1, \ldots, i-1\} \quad \implies \quad \begin{cases} \mathbb{P}\left(\operatorname{supp}(\boldsymbol{g}(\boldsymbol{x}, \xi)) \not\subset \{1, \ldots, i-1\}\right) \leq p, \\ \mathbb{P}\left(\operatorname{supp}(\boldsymbol{g}(\boldsymbol{x}, \xi)) \subset \{1, \ldots, i\}\right) = 1. \end{cases}$$

For a probability-$p$ zero-chain, at each iteration, a new coordinate is discovered with probability at most $p$ if $\operatorname{supp}(\mathsf{P}_{\mathcal{X}}(\boldsymbol{x})) = \operatorname{supp}(\boldsymbol{x})$ for all $\boldsymbol{x}$. Therefore, it takes at least $1/p$ steps in expectation to discover a new coordinate. Formally, the following lemma states that it takes $\mathcal{O}(T/p)$ iterations to reach the end of a probability-$p$ zero-chain with length $T$.

**Lemma 3** ([3, Lemma 1]). *Let $f : \mathcal{X} \to \mathbb{R}$, where $\mathcal{X} \in \mathbb{R}^T$ satisfying $\operatorname{supp}(\mathsf{P}_{\mathcal{X}}(\boldsymbol{x})) = \operatorname{supp}(\boldsymbol{x})$ for all $\boldsymbol{x} \in \mathbb{R}^T$, be a probability-$p$ zero-chain with a stochastic first-order oracle. For any first-order algorithm, we have with probability at least $1 - \delta$, $x_T^t = 0$ for all $t \leq \frac{T - \log(1/\delta)}{2p}$.*

Therefore, the gradient complexity is enlarged by a factor of $1/p$ compared with the deterministic setting. To obtain a lower bound in the stochastic setting, we first find a zero-chain $f$ satisfying the two requirements in the general strategy we presented above. Then we construct a stochastic first-order oracle which discovers the next coordinate with probability $p$ so that we obtain a probability-$p$ zero-chain. Note that we can not choose an arbitrarily small $p$ since we need to ensure the variance of the stochastic oracle is bounded.

For the ease of exposition of our construction, we now briefly review the lower bound construction for smooth strongly-convex minimization by Nesterov [37] and that for smooth nonconvex optimization by Carmon et al. [5]. As we shall see in the next section, the hard functions in these two cases are the building blocks of our construction.

## 4.1 Smooth strongly-convex minimization

Nesterov [37] constructed the following hard instance for smooth, strongly-convex functions:

$$f^{\mathrm{sc}}(\boldsymbol{x}) := \frac{\mu(\kappa-1)}{8}\left((x_1-1)^2 + \sum_{i=1}^{\infty}(x_i-x_{i+1})^2\right) + \frac{\mu}{2}\|\boldsymbol{x}\|_2^2. \tag{2}$$

Equivalently, $f^{\mathrm{sc}}(\boldsymbol{x}) = \frac{\mu(\kappa-1)}{8}(\boldsymbol{x}^\top A^{\mathrm{sc}}\boldsymbol{x} - 2x_1 + 1) + \frac{\mu}{2}\|\boldsymbol{x}\|_2^2$ for the tri-diagonal matrix $A^{\mathrm{sc}}$ given by

$$A^{\mathrm{sc}} := \begin{pmatrix} 2 & -1 & & & \\ -1 & 2 & -1 & & \\ & -1 & 2 & \ddots & \\ & & \ddots & \ddots & \ddots \end{pmatrix}. \tag{3}$$

It is straightforward to verify that $A^{\mathrm{sc}}$ is positive semi-definite and $\|A^{\mathrm{sc}}\|_2 \leq 4$. Hence, $f^{\mathrm{sc}}$ is $L$-smooth and $\mu$-strongly-convex for $L := \mu\kappa$. Importantly, if $\mathrm{supp}(\boldsymbol{x}) \subset \{1,\ldots,i-1\}$, we can verify:

$$\mathrm{supp}\left(\nabla f^{\mathrm{sc}}(\boldsymbol{x})\right) = \mathrm{supp}\left(A^{\mathrm{sc}}\boldsymbol{x}\right) \cup \mathrm{supp}\left(\boldsymbol{x}\right) \cup \{1\} \subset \{1,\ldots,i\},$$

where we use the fact that $\mathrm{supp}\left(A^{\mathrm{sc}}\boldsymbol{x}\right) \subset \{1,\ldots,i\}$ because $A^{\mathrm{sc}}$ is a tri-diagonal matrix. Hence, by Definition 6, $f^{\mathrm{sc}}$ is a zero-chain. Based on the general strategy above, it suffices to lower bound $f^{\mathrm{sc}}(\boldsymbol{x}) - f^{\mathrm{sc}}(\boldsymbol{x}^*)$ when fewer than $T$ coordinates are non-zero. Note that the minimizer $\boldsymbol{x}^*$ of $f^{\mathrm{sc}}$ is given by $x_i^* = q^i$ where $q = \frac{\sqrt{\kappa}-1}{\sqrt{\kappa}+1}$. Then with fewer than $T$ gradient oracles, for every $t \geq T$, the $t$-th coordinate of $\boldsymbol{x}$ is still zero. Therefore,

$$f^{\mathrm{sc}}(\boldsymbol{x}) - f^{\mathrm{sc}}(\boldsymbol{x}^*) \geq \frac{\mu}{2}\|\boldsymbol{x}-\boldsymbol{x}^*\|_2^2 \geq \frac{\mu}{2}\sum_{i=T}^{\infty}(x_i^*)^2 \geq \frac{\mu}{2}q^{2T}\|\boldsymbol{x}^0-\boldsymbol{x}^*\|_2^2.$$

Hence, to find a solution satisfying $f^{\mathrm{sc}}(\boldsymbol{x}) - f^{\mathrm{sc}}(\boldsymbol{x}^*) \leq \epsilon$, we need gradient complexity $T \geq \widetilde{\Omega}(\sqrt{\kappa})$.

## 4.2 Smooth nonconvex minimization

Another key component in our proof is to lower bound the complexity of nonconvex functions. Part of our construction is derived from Carmon et al. [5] who constructed the following unscaled function $\bar{f}^{\mathrm{nc}} : \mathbb{R}^T \to \mathbb{R}$:

$$\bar{f}^{\mathrm{nc}}(\boldsymbol{x}) := -\Psi(1)\Phi(x_1) + \sum_{i=2}^{T}\left[\Psi(-x_{i-1})\Phi(-x_i) - \Psi(x_{i-1})\Phi(x_i)\right], \tag{4}$$

where the component functions are

$$\Psi(x) := \begin{cases} 0 & x \leq 1/2 \\ \exp\left(1 - \frac{1}{(2x-1)^2}\right) & x > 1/2 \end{cases} \quad \text{and} \quad \Phi(x) := \sqrt{e}\int_{-\infty}^{x} e^{-\frac{1}{2}t^2}\,dt.$$

The above function also has the zero-chain property defined in 6. Following the general strategy discussed in the previous subsection, by appropriately rescaling $\bar{f}^{\mathrm{nc}}$ so that it meets the requirement of the function class of interest, Carmon et al. [5] derived a lower bound of $T_{\mathrm{nc}} := \Omega\left(1/\epsilon^2\right)$ gradient oracles. More details can be found in Appendix A.

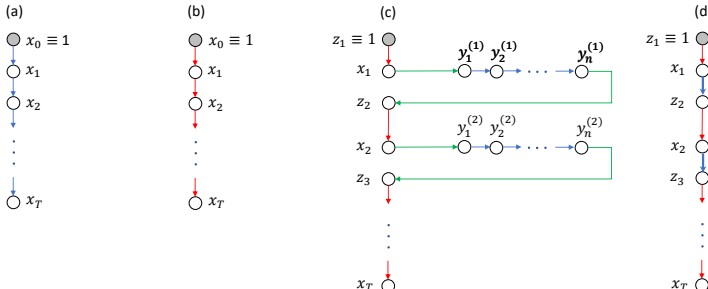

Figure 1: Chains (a, b, c, d) correspond to $f^{\text{sc}}, \bar{f}^{\text{nc}}, \bar{f}^{\text{nc-sc}}, \bar{f}_m^{\text{nc-sc}}$, respectively. Arrows in different colors represent different types of connections between two variables. For example, a blue arrow from $u$ to $v$ corresponds to a term with $(u - v)^2$ in the function. A red arrow corresponds to $\Psi(-u)\Phi(-v) - \Psi(u)\Phi(v)$. Green and bold blue arrows are quadratic connections that need to be carefully designed. The effective length of a bold blue arrow is $n$ and that of all other arrows is 1.

## 5 Our construction

The hard examples discussed in the previous section show that one can easily construct an additive lower bound by making the objective as the sum of two functions $f^{\text{sc}}$ and $\bar{f}^{\text{nc}}$. However, if we would like to improve the lower bound by a multiplicative factor in the condition number, it is far from obvious how one should compose $f^{\text{sc}}$ with $\bar{f}^{\text{nc}}$. We describe our approach in the subsections below.

### 5.1 Construction of the hard instance in the deterministic setting

Before discussing the details of our construction to prove Theorem 1, we first highlight the main difficulty and our approach at a high level to provide insight and intuition. First, we need to pinpoint the main difficulty.

**The main difficulty.** Given the aforementioned lower bound strategy, we need to find a zero-chain $\bar{f}^{\text{nc-sc}} \in \mathcal{F}(L, \mu, \Delta)$ as our hard instance. A natural way to construct $\bar{f}^{\text{nc-sc}}$ is to combine the ideas from the hard instances in [37] and [5]. The main challenge is how to find a good way of combination such that the two components do not interfere with each other's essential properties and that their strengths can be exploited multiplicatively to contribute to the lower bound.

**The key idea.** The novel combination structure we propose is illustrated in Figure 1. The body of the chain is like $\bar{f}^{\text{nc}}$ (4), which contains the minimization variables $\boldsymbol{x}, \boldsymbol{z}$. Then we insert $T - 1$ $f^{\text{sc}}$-style (2) sub-chains with length $n$, which contain the maximization variable $\boldsymbol{y}$, into the main chain in the way shown in Figure 1(c). Figure 1(d) shows the function $\bar{f}_m^{\text{nc-sc}}(\cdot) := \max_{\boldsymbol{y}} \bar{f}^{\text{nc-sc}}(\cdot; \boldsymbol{y})$ obtained by maximizing $\boldsymbol{y}$. Note that the effective length of chain (d) is $\mathcal{O}(nT) = \mathcal{O}(\sqrt{\kappa}T)$, for the largest allowed choice of $n = \mathcal{O}(\sqrt{\kappa})$. Therefore, supposing we can show that $\bar{f}_m^{\text{nc-sc}}$ shares similar properties to $\bar{f}^{\text{nc}}$ used in [5], we obtain a lower bound of $\Omega\left(\sqrt{\kappa}T_{\text{nc}}\right) = \Omega\left(\sqrt{\kappa}/\epsilon^2\right)$.

However, the requirement that $\bar{f}_m^{\text{nc-sc}}$ behaves similarly to $\bar{f}^{\text{nc}}$ poses another challenge as it basically means that the red arrows in chain (d) dominate the bold blue ones, although their numbers are roughly equal. To overcome this challenge, we carefully design the green arrows directly and the bold blue arrows indirectly. We also need to restrict $n$ such that $n = \mathcal{O}(\sqrt{\kappa})$.

**Construction of the hard instance.** The formal expression of $\bar{f}^{\text{nc-sc}} : (\mathbb{R}^T \times \mathbb{R}^{T-1}) \times \mathbb{R}^{n(T-1)} \to \mathbb{R}$ is given by

$$
\bar{f}^{\text{nc-sc}}(\boldsymbol{x}, \boldsymbol{z}; \bar{\boldsymbol{y}}) = -\Psi(1)\Phi(x_1) + \sum_{i=2}^{T} \left[\Psi(-z_i)\Phi(-x_i) - \Psi(z_i)\Phi(x_i)\right]
$$
$$
+ \sum_{i=1}^{T-1} h(x_i, z_{i+1}; \bar{\boldsymbol{y}}^{(i)}) + \sum_{i=1}^{T-1} \left[c_1 x_i^2 + c_2 z_{i+1}^2\right], \quad (5)
$$

where $\boldsymbol{x}, \boldsymbol{z}$ are the minimization variables and $\bar{\boldsymbol{y}} = [\bar{\boldsymbol{y}}^{(1)}, \ldots, \bar{\boldsymbol{y}}^{(T-1)}]$, where each $\bar{\boldsymbol{y}}^{(i)} \in \mathbb{R}^n$, is the maximization variable. The last term of (5) indirectly affects the bold blue arrows in Figure 1(d), where $c_1$ and $c_2$ are some parameters (not necessarily nonnegative) we choose later to obtain a much simpler expression to analyze. The function $h : (\mathbb{R} \times \mathbb{R}) \times \mathbb{R}^n \to \mathbb{R}$ is defined as

$$h(x, z; \boldsymbol{y}) := -\frac{1}{2} \sum_{i=1}^{n-1} (y_i - y_{i+1})^2 - \frac{1}{2n^2} \|\boldsymbol{y}\|_2^2 + \sqrt{\frac{C}{n}} \left( xy_1 - \frac{1}{2} zy_n \right)$$

$$= -\frac{1}{2} \boldsymbol{y}^\top \left( \frac{1}{n^2} I_n + A \right) \boldsymbol{y} + \sqrt{\frac{C}{n}} \boldsymbol{b}_{x,z}^\top \boldsymbol{y}, \tag{6}$$

where $C > 0$ is a large enough numerical constant specified later and $\boldsymbol{b}_{x,z} := x\boldsymbol{e}_1 - \frac{1}{2} z\boldsymbol{e}_n$. Here the term $\sqrt{\frac{C}{n}} \boldsymbol{b}_{x,z}^\top \boldsymbol{y}$ characterizes the green arrows in Figure 1, where the $\sqrt{\frac{C}{n}}$ factor ensures $h_m(x, z) := \max_{\boldsymbol{y} \in \mathbb{R}^n} h(x, z; \boldsymbol{y})$ has an $\mathcal{O}(1)$ dependence on $n$ as shown in Lemma 4. The matrix $A \in \mathbb{R}^{n \times n}$ is defined as

$$A := \begin{pmatrix} 1 & -1 & & & \\ -1 & 2 & -1 & & \\ & -1 & \ddots & \ddots & \\ & & \ddots & 2 & -1 \\ & & & -1 & 1 \end{pmatrix}, \tag{7}$$

which is the finite version of (3) in [37]'s instance except that its $(1,1)$-th and $(n,n)$-th entries are 1. This change is necessary in our example. Note that for fixed scalar variables $x$ and $z$, $h(x, z; \cdot)$ is $\frac{1}{n^2}$-strongly-concave. Also, $\bar{f}^{\text{nc-sc}}$ is $\Omega(1)$-smooth (Lemma 9). Therefore, $\kappa = \Omega(n^2)$, implying we should choose $n = \mathcal{O}(\sqrt{\kappa})$. We can compute

$$h_m(x, z) := \max_{\boldsymbol{y} \in \mathbb{R}^n} h(x, z; \boldsymbol{y}) = \frac{C}{2n} \boldsymbol{b}_{x,z}^\top \left( \frac{1}{n^2} I_n + A \right)^{-1} \boldsymbol{b}_{x,z}. \tag{8}$$

The following lemma shows that $h_m$ actually has a much simpler expression.

**Lemma 4.** *Suppose $n \geq 10$, we have*

$$h_m(x, z) = C \left( \frac{a_1}{2} x^2 - \frac{1}{2} a_2 xz + \frac{a_1}{8} z^2 \right), \tag{9}$$

*where $a_1, a_2 > 0$ are "almost" numerical constants. That is to say, although $a_1, a_2$ depends on $n$, we have $0 < d_1 \leq a_1 \leq f_1$ and $0 < d_2 \leq a_2 \leq f_2$ and $d_1, d_2, f_1, f_2$ are numerical constants.*

Choosing $c_1 = C(a_2 - a_1)/2$, $c_2 = C(a_2 - a_1)/8$ and $C = 12/a_2$, we obtain

$$\bar{f}_m^{\text{nc-sc}}(\boldsymbol{x}, \boldsymbol{z}) := \max_{\bar{\boldsymbol{y}} \in \mathbb{R}^{n(T-1)}} \bar{f}^{\text{nc-sc}}(\boldsymbol{x}, \boldsymbol{z}; \bar{\boldsymbol{y}})$$

$$= -\Psi(1)\Phi(x_1) + \sum_{i=2}^{T} [\Psi(-z_i)\Phi(-x_i) - \Psi(z_i)\Phi(x_i)] + 6\sum_{i=1}^{T-1} \left( x_i - \frac{1}{2} z_{i+1} \right)^2.$$

We show in the appendix that $\bar{f}_m^{\text{nc-sc}}$ shares similar properties to $\bar{f}^{\text{nc}}$. Then we can prove Theorem 1 by appropriately rescaling $\bar{f}_m^{\text{nc-sc}}$ in the same way as done in [5]. The detailed proof is deferred in Appendix C.

## 5.2 Construction of the hard instance in the stochastic setting

We start this section by discussing why the techniques in [3] do not directly apply to our construction in the deterministic setting. The following lemma from [3] shows how to construct a probability-$p$ zero-chain by constructing a stochastic first-order oracle over a given zero-chain.

**Lemma 5** ([3, Lemma 3]). *Let $f : \mathcal{X} \to \mathbb{R}$ be a zero-chain on $\mathcal{X} \subset \mathbb{R}^T$. For $\boldsymbol{x} \in \mathcal{X}$, let $i^*(\boldsymbol{x}) := \inf\{i \in [T] : x_i = 0\}$ be the next coordinate to discover. For $p \in (0, 1]$, define*

$$[\boldsymbol{g}(\boldsymbol{x}, \xi)]_i := \begin{cases} \frac{\xi}{p} \nabla_i f(\boldsymbol{x}) & \text{if } i = i^*(\boldsymbol{x}) \\ \nabla_i f(\boldsymbol{x}) & \text{otherwise,} \end{cases}$$

*where $\xi \sim \text{Bernoulli}(p)$. Suppose $\|\nabla f(\boldsymbol{x})\|_\infty \le G$ for all $\boldsymbol{x} \in \mathcal{X}$. Then $O : \boldsymbol{x} \mapsto (f(\boldsymbol{x}, \boldsymbol{g}(\boldsymbol{x}, \xi))$ is a stochastic first-order oracle with bounded variance $\sigma^2 \le G^2(1-p)/p$. Also, $f$ equipped with $O$ is a probability-$p$ zero-chain.*

Therefore, a tempting way to obtain our hard instance in the stochastic setting is to construct a probability-$p$ zero-chain as in Lemma 5 directly over $\bar{f}^{\text{nc-sc}}$. However, as suggested by Lemma 5, to ensure the variance of the oracle is bounded, we require $\left\|\nabla \bar{f}^{\text{nc-sc}}(\boldsymbol{x}, \boldsymbol{z}; \bar{\boldsymbol{y}})\right\|_\infty \le G$ for some bounded $G < \infty$. Actually we need $G = \mathcal{O}(1)$ to obtain a nontrivial lower bound. However, $\bar{f}^{\text{nc-sc}}$ has an unconstrained quadratic component whose gradient is unbounded over the whole space. Therefore, we have to restrict its domain to be a bounded hypercube. However, it turns out to be impossible to find any hypercube such that $\left\|\nabla \bar{f}^{\text{nc-sc}}(\boldsymbol{x}, \boldsymbol{z}; \bar{\boldsymbol{y}})\right\|_\infty \le \mathcal{O}(1)$ without losing the properties of $\bar{f}_m^{\text{nc-sc}}$. To overcome this difficulty, we need to carefully modify the quadratic components in $\bar{f}^{\text{nc-sc}}$, i.e., the blue and green arrows in Figure 1(c).

Formally, define the hypercube $\mathcal{C}_R^d \subset \mathbb{R}^d$ as

$$\mathcal{C}_R^d = \{\boldsymbol{x} \in \mathbb{R}^d \mid \|\boldsymbol{x}\|_\infty \le R\}.$$

Our hard instance in the stochastic setting, $\bar{f}^{\text{nc-sc-sg}} : (\mathcal{C}_{R_1}^T \times \mathcal{C}_{R_1}^{T-1}) \times \mathcal{C}_{nR_2}^{n(T-1)} \to \mathbb{R}$, where $R_1$ and $R_2$ are positive numerical constants to be set later, is given by

$$\bar{f}^{\text{nc-sc-sg}}(\boldsymbol{x}, \boldsymbol{z}; \bar{\boldsymbol{y}}) = -\Psi(1)\Phi(x_1) + \sum_{i=2}^{T} [\Psi(-z_i)\Phi(-x_i) - \Psi(z_i)\Phi(x_i)]$$

$$+ \sum_{i=1}^{T-1} h^{\text{sg}}(x_i, z_{i+1}; \bar{\boldsymbol{y}}^{(i)}) + \sum_{i=1}^{T-1} \left[c_1 x_i^2 + c_2 z_{i+1}^2\right], \quad (10)$$

where

$$h^{\text{sg}}(x, z; \boldsymbol{y}) := \frac{C}{n} \left[ -\frac{1}{2} \boldsymbol{y}^\top \left( \frac{1}{n^2} I_n + A \right) \boldsymbol{y} + \boldsymbol{b}_{x,z}^\top \boldsymbol{y} \right]. \quad (11)$$

Here $C, c_1, c_2, \boldsymbol{b}_{x,z}$ are the same as those in (5).

By verifying that $\left\|\nabla \bar{f}^{\text{nc-sc-sg}}(\boldsymbol{x}, \boldsymbol{z}; \bar{\boldsymbol{y}})\right\|_\infty \le \mathcal{O}(1)$ (Lemma 11.*iv*), we are able to construct a probability-$p$ zero-chain as in Lemma 5 over the rescaled version of $\bar{f}^{\text{nc-sc}}$. According to Lemma 5, to ensure the variance of the rescaled stochastic oracle is bounded by $\sigma^2$, $p$ has to be at least $\Omega(\epsilon^2/\sigma^2)$. Then we obtain a lower bound for the stochastic setting of

$$\Omega\left(\frac{nT_{\text{nc}}}{p}\right) = \Omega\left(\frac{\kappa^{1/3}}{\epsilon^4}\right).$$

Note that the deterministic lower bound is $\Omega(\frac{\sqrt{\kappa}}{\epsilon^2})$ which is a special case of the stochastic setting. Therefore we derive a lower bound of

$$\Omega\left(\max\left\{\frac{\sqrt{\kappa}}{\epsilon^2}, \frac{\kappa^{1/3}\sigma^2}{\epsilon^4}\right\}\right) = \Omega\left(\frac{\sqrt{\kappa}}{\epsilon^2} + \frac{\kappa^{1/3}}{\epsilon^4}\right).$$

Detailed analyses are deferred to Appendix D.

## 6 Conclusion and discussion

In this paper, we proved lower bounds on both the deterministic and the stochastic oracle complexities of nonconvex-strongly-concave min-max optimization for first-order zero-respecting algorithms. Our lower bound in the deterministic setting matches the existing upper bound [27] up to log factors. However, there is still a gap between our lower bound and the upper bound in [26] in the stochastic setting. How to close this gap is an open question. Apart from this, several other questions are worth consideration.

First, one immediate next step is to check if the proposed lower bounds hold for arbitrary, potentially randomized algorithms. We believe that the results are likely to hold but may introduce unexpected

complications. Second, so far we have focused on nonconvex-strongly-concave min-max optimization. However, it remains open what the tight lower bound is in the more general nonconvex-concave min-max optimization. Moreover, what about nonconvex-nonconcave min-max optimization? To answer this question, a good measure of the suboptimality is a prerequisite. Last but not least, we only consider first-order oracles. It is also interesting to obtain a lower bound for functions with higher-order smoothness and oracles.

## 7 Funding transparency statement

Funding in direct support of this work: ONR grant N00014-20-1-2394, MIT-IBM Watson grant, NSF BIGDATA grant 1741341, and IIIS young scholar fellowship.

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
