## A  Properties of the nonconvex lower bound example (4)

We enumerate all relevant properties of $\Phi$ and $\Psi$ used in the analysis in the following lemma.

**Lemma 6** ([5, Lemma 1]). *The functions $\Phi$ and $\Psi$ satisfy*

  *i. For all $x \leq 1/2$ and $k \in \mathbb{N}$, $\Psi^{(k)}(x) = 0$.*

 *ii. For all $x \geq 1$ and $|y| < 1$, $\Psi(x)\Phi'(y) > 1$.*

*iii. Both $\Psi$ and $\Phi$ are infinitely differentiable. For all $k \in \mathbb{N}$, we have*

$$\sup_x |\Psi^{(k)}(x)| \leq \exp\left(\frac{5k}{2}\log(4k)\right) \quad \text{and} \quad \sup_x |\Phi^{(k)}(x)| \leq \exp\left(\frac{3k}{2}\log\frac{3k}{2}\right).$$

*iv. The functions and derivatives $\Psi$, $\Psi'$, $\Phi$, $\Phi'$ are non-negative and bounded, with*

$$0 < \Psi < e, 0 < \Psi' < \sqrt{54/e}, 0 < \Phi < \sqrt{2\pi e}, 0 < \Phi' < \sqrt{e}.$$

Note that $\Psi(0) = \Psi'(0) = 0$ by Lemma 6.*i*. Then it is easy to verify that $\frac{\partial \bar{f}^{\mathrm{nc}}(\boldsymbol{x})}{\partial x_i} = 0$ if $x_i = x_{i-1} = 0$. Therefore, if $\mathrm{supp}(\boldsymbol{x}) \subset \{1, \ldots, i-1\}$, i.e., $x_j = 0$ for all $j \geq i$, we have $\frac{\partial \bar{f}^{\mathrm{nc}}(\boldsymbol{x})}{\partial x_j} = 0$ for all $j \geq i+1$. Hence, $\mathrm{supp}(\nabla \bar{f}^{\mathrm{nc}}) \subset \{1, \ldots, i\}$, which implies $\bar{f}^{\mathrm{nc}}$ is a zero-chain. Define $x_0 \equiv 1$ for simplicity. As long as the algorithm has not reached the end of the chain, there must be a phase transition point $1 \leq k \leq T$ such that $|x_k| < 1$ and $|x_{k-1}| \geq 1$. Using Lemma 6.*ii*, one can bound $\left\|\nabla \bar{f}^{\mathrm{nc}}(\boldsymbol{x}^t)\right\|_2 \geq \left|\frac{\partial \bar{f}^{\mathrm{nc}}(\boldsymbol{x})}{\partial x_k}\right| > 1$. By appropriately rescaling $\bar{f}^{\mathrm{nc}}$ so that it meets the requirement of the function class of interest, Carmon et al. [5] derived a lower bound of $T_{\mathrm{nc}} := \Omega\left(1/\epsilon^2\right)$ gradient oracles.

## B  A Useful lemma

We first present a lemma useful for analyzing the quadratic components in our examples.

**Lemma 7.** *Denote $\alpha = \frac{1}{n^2}$ and let $B = (\alpha I_n + A)^{-1}$ where $A$ is the matrix defined in (7). If $n \geq 10$, we have for all $1 \leq i \leq n$,*

$$0.1n \leq B_{i,1} \leq 20n.$$

*Proof of Lemma 7.* Let $M$ be the cofactor matrix of $\alpha I_n + A$. We have

$$B = \frac{M^\top}{\det(\alpha I_n + A)}.$$

So we only need to compute $\det(\alpha I_n + A)$ and $M_{1,i}$ for all $1 \leq i \leq n$. Note that all of them are determinants of tridiagonal matrices which can be computed using a three-term recurrence relation [12]. Let

$$p = 1 + \frac{\alpha}{2} + \sqrt{\alpha + \frac{\alpha^2}{4}}, \quad q = 1 + \frac{\alpha}{2} - \sqrt{\alpha + \frac{\alpha^2}{4}}$$

be the solutions of the following characteristic equation

$$x^2 - (2+\alpha)x + 1 = 0.$$

By standard calculations, we have

$$\det(\alpha I_n + A) = \frac{\left(\alpha + \frac{\alpha^2}{2}\right)\left(p^{n-1} - q^{n-1}\right) + \alpha\sqrt{\alpha + \frac{\alpha^2}{4}}\left(p^{n-1} + q^{n-1}\right)}{2\sqrt{\alpha + \frac{\alpha^2}{4}}},$$

$$M_{1,i} = \frac{\frac{\alpha}{2}\left(p^{n-i} - q^{n-i}\right) + \sqrt{\alpha + \frac{\alpha^2}{4}}\left(p^{n-i} + q^{n-i}\right)}{2\sqrt{\alpha + \frac{\alpha^2}{4}}}.$$

Define $D = p^{n-1}$, $E = D - \frac{1}{D}$, and $F = D + \frac{1}{D}$. We have

$$0 \le p^{n-i} - q^{n-i} \le E \text{ and } 2 \le p^{n-i} + q^{n-i} \le F.$$

Therefore

$$\det(\alpha I_n + A) = \frac{\left(\alpha + \frac{\alpha^2}{2}\right) E + \alpha\sqrt{\alpha + \frac{\alpha^2}{4}} F}{2\sqrt{\alpha + \frac{\alpha^2}{4}}},$$

$$1 \le M_{1,n} \le M_{1,i} \le M_{1,1} = \frac{\frac{\alpha}{2} E + \sqrt{\alpha + \frac{\alpha^2}{4}} F}{2\sqrt{\alpha + \frac{\alpha^2}{4}}}.$$

Noting $\alpha = \frac{1}{n^2}$, we have

$$D = p^{n-1} = \left(1 + \frac{1}{2n^2} + \frac{1}{n}\sqrt{1 + \frac{1}{4n^2}}\right)^{n-1}.$$

We can bound $2 \le D \le 8$ if $n \ge 10$. Then it is straightforward to upper and lower bound $\det(\alpha I_n + A)$ and $M_{1,i}$ and then obtain the bound of $B_{i,1}$. If $n \ge 10$, we have

$$0.1n \le B_{i,1} \le 20n, \forall 1 \le i \le n.$$

$\square$

## C  Proofs for the lower bound in the deterministic setting

*Proof of Lemma 4.* Let $B = \left(\frac{1}{n^2} I_n + A\right)^{-1}$ where $A$ is the matrix defined in (7). By symmetry, we have $B_{1,1} = B_{n,n}$ and $B_{1,n} = B_{n,1}$. Then we have

$$h_m(x, z) = \frac{C}{2n}\left(B_{1,1} x^2 - B_{n,1} xz + \frac{B_{1,1}}{4} z^2\right).$$

Let $a_1 = B_{1,1}/n$ and $a_2 = B_{n,1}/n$. By Lemma 7 we know $0.1 \le a_1, a_2 \le 20$ and complete the proof. $\square$

To prove the main theorem, we need several additional lemmas. The following lemma gives a lower bound of the gradient norm when the algorithm hasn't reached the end of the chain.

**Lemma 8.** *If $|z_i| < 1$ for some $i \le T$, then $\left\|\nabla \bar{f}_m^{nc\text{-}sc}(\boldsymbol{x}, \boldsymbol{z})\right\|_2 > \frac{1}{3}$.*

*Proof of Lemma 8.* We define $z_1 \equiv 1$ for simplicity. Since $|z_i| < 1$ and $|z_1| \ge 1$, we are able to find some $1 < j \le i$ to be the smallest $j$ for which $|z_j| < 1$. So we know $|z_{j-1}| \ge 1$. We can compute

$$\frac{\partial \bar{f}_m^{nc\text{-}sc}(\boldsymbol{x}, \boldsymbol{z})}{\partial x_{j-1}} = -\Psi(-z_{j-1})\Phi'(-x_{j-1}) - \Psi(z_{j-1})\Phi'(x_{j-1}) + 12\left(x_{j-1} - \frac{1}{2} z_j\right)$$

$$=: p(x_{j-1}, z_{j-1}) + 12\left(x_{j-1} - \frac{1}{2} z_j\right),$$

$$\frac{\partial \bar{f}_m^{nc\text{-}sc}(\boldsymbol{x}, \boldsymbol{z})}{\partial z_j} = -\Psi'(-z_j)\Phi(-x_j) - \Psi'(z_j)\Phi(x_j) - 6\left(x_{j-1} - \frac{1}{2} z_j\right)$$

$$=: q(x_j, z_j) - 6\left(x_{j-1} - \frac{1}{2} z_j\right).$$

Note that Lemma 6.*iv* implies for all $2 \le i \le T$,

$$-5 < p(x_j, z_j) < 0, \quad -20 < q(x_j, z_j) < 0.$$

There are two possible cases

1. If $|x_{j-1}| < 1$, we have $p(x_{j-1}, z_{j-1}) < -1$ by Lemma 6.*ii*. Then

$$\frac{\partial \bar{f}_m^{\text{nc-sc}}(\boldsymbol{x}, \boldsymbol{z})}{\partial x_{j-1}} + 2 \cdot \frac{\partial \bar{f}_m^{\text{nc-sc}}(\boldsymbol{x}, \boldsymbol{z})}{\partial z_j} = p(x_{j-1}, z_{j-1}) + 2q(x_j, z_j) < -1.$$

Therefore we can bound

$$\left\| \nabla \bar{f}_m^{\text{nc-sc}}(\boldsymbol{x}, \boldsymbol{z}) \right\|_2 \geq \max \left\{ \left| \frac{\partial \bar{f}_m^{\text{nc-sc}}(\boldsymbol{x}, \boldsymbol{z})}{\partial x_{j-1}} \right|, \left| \frac{\partial \bar{f}_m^{\text{nc-sc}}(\boldsymbol{x}, \boldsymbol{z})}{\partial z_j} \right| \right\} > \frac{1}{3}.$$

2. Otherwise if $|x_{j-1}| \geq 1$, we have $12 \left| x_{j-1} - \frac{1}{2} z_j \right| > 6$. Since $|p(x_{j-1}, z_{j-1})| < 5$, we must have

$$\left\| \nabla \bar{f}_m^{\text{nc-sc}}(\boldsymbol{x}, \boldsymbol{z}) \right\|_2 \geq \left| \frac{\partial \bar{f}_m^{\text{nc-sc}}(\boldsymbol{x}, \boldsymbol{z})}{\partial x_{j-1}} \right| > 1.$$

$\square$

Now we verify the smoothness and boundedness requirements of the function class we consider.

**Lemma 9.** $\bar{f}^{\text{nc-sc}}$ and $\bar{f}_m^{\text{nc-sc}}$ satisfy the following.

 *i.* $\bar{f}_m^{\text{nc-sc}}(\boldsymbol{0}, \boldsymbol{0}) - \inf_{\boldsymbol{x} \in \mathbb{R}^T, \boldsymbol{z} \in \mathbb{R}^{T-1}} \bar{f}_m^{\text{nc-sc}}(\boldsymbol{x}, \boldsymbol{z}) \leq 12T$.

 *ii.* $\bar{f}^{\text{nc-sc}}$ is $\ell_0$-smooth for some numerical constant $\ell_0$.

*Proof of Lemma 9.*

 i. First note that $\bar{f}_m^{\text{nc-sc}}(\boldsymbol{0}, \boldsymbol{0}) = -\Phi(1)\Phi(0) \leq 0$. Also, by Lemma 6.*iv*, we have for all $\boldsymbol{x} \in \mathbb{R}^T, \boldsymbol{z} \in \mathbb{R}^{T-1}$,

$$\bar{f}_m^{\text{nc-sc}}(\boldsymbol{x}, \boldsymbol{z}) \geq -\Psi(1)\Phi(x_1) - \sum_{i=2}^{T} \Psi(z_i)\Phi(x_i) \geq -12T.$$

 Therefore $\bar{f}_m^{\text{nc-sc}}(\boldsymbol{0}, \boldsymbol{0}) - \inf_{\boldsymbol{x} \in \mathbb{R}^T, \boldsymbol{z} \in \mathbb{R}^{T-1}} \bar{f}_m^{\text{nc-sc}}(\boldsymbol{x}, \boldsymbol{z}) \leq 12T$.

 ii. Let $\boldsymbol{v} = (\boldsymbol{x}, \boldsymbol{z}, \bar{\boldsymbol{y}})$ be the variable of $\bar{f}^{\text{nc-sc}}$. We know $\frac{\partial \bar{f}^{\text{nc-sc}}}{\partial v_i \partial v_j} \neq 0$ only if $i = j$ or $v_i$ and $v_j$ are directly connected in the chain shown in Figure 1 (c). Therefore the Hessian of $\bar{f}^{\text{nc-sc}}$ is tridiagonal if we rearranging the coordinates of $\boldsymbol{v}$ according to the order of the chain. By Lemma 6.*iii* and the expression of $\bar{f}^{\text{nc-sc}}$, it is straightforward to verify that each tridiagonal entry of the Hessian is $\mathcal{O}(1)$. Therefore the $\ell_2$ norm of the Hessian is $\mathcal{O}(1)$, which means $\bar{f}^{\text{nc-sc}}$ is $\mathcal{O}(1)$-smooth.

$\square$

With all the above properties of $\bar{f}^{\text{nc-sc}}$ and $\bar{f}_m^{\text{nc-sc}}$, we are ready to show Theorem 1.

*Proof of Theorem 1.* As in [5], we construct the hard instance $f^{\text{nc-sc}}$ by appropriately rescaling $\bar{f}^{\text{nc-sc}}$ defined in (5),

$$f^{\text{nc-sc}}(\boldsymbol{x}, \boldsymbol{z}; \bar{\boldsymbol{y}}) = \frac{L\lambda^2}{\ell_0} \bar{f}^{\text{nc-sc}}\left( \frac{\boldsymbol{x}}{\lambda}, \frac{\boldsymbol{z}}{\lambda}; \frac{\bar{\boldsymbol{y}}}{\lambda} \right),$$

where $\lambda > 0$ is some parameter to be determined later and $\ell_0$ is the smoothness parameter defined in Lemma 9.*ii*. Note that we can show

$$f_m^{\text{nc-sc}}(\boldsymbol{x}, \boldsymbol{z}) := \max_{\bar{\boldsymbol{y}} \in \mathbb{R}^{n(T-1)}} f^{\text{nc-sc}}(\boldsymbol{x}, \boldsymbol{z}; \bar{\boldsymbol{y}}) = \max_{\boldsymbol{u} \in \mathbb{R}^{n(T-1)}} \frac{L\lambda^2}{\ell_0} \bar{f}^{\text{nc-sc}}\left( \frac{\boldsymbol{x}}{\lambda}, \frac{\boldsymbol{z}}{\lambda}; \boldsymbol{u} \right) = \frac{L\lambda^2}{\ell_0} \bar{f}_m^{\text{nc-sc}}\left( \frac{\boldsymbol{x}}{\lambda}, \frac{\boldsymbol{z}}{\lambda} \right),$$

which means the order of maximization and rescaling can be interchanged. After the rescaling, $f^{\text{nc-sc}}$ is still a zero-chain. Also, if $z_T = 0$ for some $(\boldsymbol{x}, \boldsymbol{z}; \bar{\boldsymbol{y}})$, Lemma 8 shows that

$$\left\| \nabla \bar{f}_m^{\text{nc-sc}} \left( \frac{\boldsymbol{x}}{\lambda}, \frac{\boldsymbol{z}}{\lambda} \right) \right\|_2 > \frac{1}{3}.$$

Therefore

$$\left\| \nabla f_m^{\text{nc-sc}} (\boldsymbol{x}, \boldsymbol{z}) \right\|_2 = \frac{L\lambda}{\ell_0} \left\| \nabla \bar{f}_m^{\text{nc-sc}} \left( \frac{\boldsymbol{x}}{\lambda}, \frac{\boldsymbol{z}}{\lambda} \right) \right\|_2 > \frac{L\lambda}{3\ell_0}.$$

Choosing $\lambda = \frac{3\ell_0 \epsilon}{L}$ garautees $\left\| \nabla f_m^{\text{nc-sc}} (\boldsymbol{x}, \boldsymbol{z}) \right\|_2 > \epsilon$.

Now we check $f^{\text{nc-sc}} \in \mathcal{F}(L, \mu, \Delta)$. Note that

$$\nabla^2 f^{\text{nc-sc}} (\boldsymbol{x}, \boldsymbol{z}; \bar{\boldsymbol{y}}) = \frac{L}{\ell_0} \nabla^2 \bar{f}^{\text{nc-sc}} \left( \frac{\boldsymbol{x}}{\lambda}, \frac{\boldsymbol{z}}{\lambda}; \frac{\bar{\boldsymbol{y}}}{\lambda} \right).$$

Therefore we know the smoothness parameter of $f^{\text{nc-sc}}$ is $L$ and the strong concavity parameter is $\frac{L}{\ell_0 n^2}$. Therefore we should choose

$$n = \left\lfloor \sqrt{\frac{L}{\mu \ell_0}} \right\rfloor$$

to make $f^{\text{nc-sc}}$ $\mu$-strongly concave in $\bar{\boldsymbol{y}}$.

Then it suffices to verify $f_m^{\text{nc-sc}}(\boldsymbol{0}, \boldsymbol{0}) - \inf_{\boldsymbol{x}, \boldsymbol{z}} f_m^{\text{nc-sc}}(\boldsymbol{x}, \boldsymbol{z}) \le \Delta$. By Lemma 9,

$$f_m^{\text{nc-sc}}(\boldsymbol{0}, \boldsymbol{0}) - \inf_{\boldsymbol{x}, \boldsymbol{z}} f_m^{\text{nc-sc}}(\boldsymbol{x}, \boldsymbol{z}) = \frac{L\lambda^2}{\ell_0} \left( \bar{f}_m^{\text{nc-sc}}(\boldsymbol{0}, \boldsymbol{0}) - \inf_{\boldsymbol{x}, \boldsymbol{z}} \bar{f}_m^{\text{nc-sc}}(\boldsymbol{x}, \boldsymbol{z}) \right) \le \frac{12 L T \lambda^2}{\ell_0},$$

which is less than $\Delta$ if choosing

$$T = \left\lfloor \frac{\ell_0 \Delta}{12 L \lambda^2} \right\rfloor = \left\lfloor \frac{L\Delta}{108 \ell_0 \epsilon^2} \right\rfloor.$$

Since $z_T^t = 0$ if $t \le n(T-1)$, we conclude that $\left\| \nabla f_m^{\text{nc-sc}}(\boldsymbol{x}^t, \boldsymbol{z}^t) \right\|_2 > \epsilon$ whenever

$$t \le n(T-1) = \frac{c_0 L \Delta \sqrt{\kappa}}{\epsilon^2}$$

for some numerical constant $c_0$. □

## D    Proofs for the lower bound in the stochastic setting

**Lemma 10.** *Let* $h_m^{sg}(x, z) := \max_{\boldsymbol{y} \in \mathcal{C}_{nR_2}^n} h^{sg}(x, z; \boldsymbol{y})$. *If* $R_2 \ge 30 R_1$, *for every* $x, z$ *such that* $|x|, |z| \le R_1$, *we have*

$$h_m^{sg}(x, z) = h_m(x, z),$$

*where* $h_m$ *is the quadratic function defined in* (8).

*Proof of Lemma 10.* Note that

$$\max_{\boldsymbol{y} \in \mathbb{R}^n} h^{sg}(x, z; \boldsymbol{y}) = \frac{C}{2n} \boldsymbol{b}_{x,z}^\top \left( \frac{1}{n^2} I_n + A \right)^{-1} \boldsymbol{b}_{x,z} = h_m(x, z).$$

It suffices to verify that

$$\max_{\boldsymbol{y} \in \mathcal{C}_{nR_2}^n} h^{sg}(x, z; \boldsymbol{y}) = \max_{\boldsymbol{y} \in \mathbb{R}^n} h^{sg}(x, z; \boldsymbol{y}),$$

i.e.,

$$\boldsymbol{y}^*(x, z) := \operatorname*{argmax}_{\boldsymbol{y} \in \mathbb{R}^n} h^{sg}(x, z; \boldsymbol{y}) \in \mathcal{C}_{nR_2}^n.$$

We can compute that

$$\boldsymbol{y}^*(x, z) = \left(\frac{1}{n^2}I_n + A\right)^{-1} \boldsymbol{b}_{x,z} = B \cdot \boldsymbol{b}_{x,z},$$

where $B = \left(\frac{1}{n^2}I_n + A\right)^{-1}$ is the matrix defined in Lemma 7. Let $y_i^*(x, z)$ be the $i$-th coordinate of $\boldsymbol{y}^*(x, z)$ for some $1 \le i \le n$. By symmetry of $B$ and Lemma 7, we have

$$
\begin{aligned}
|y_i^*(x, z)| &= \left| xB_{i,1} - \frac{1}{2}zB_{i,n} \right| \\
&= \left| xB_{i,1} - \frac{1}{2}zB_{n-i,1} \right| \\
&\le 30nR_1 \le nR_2.
\end{aligned}
$$

Therefore $\boldsymbol{y}^*(x, z) \in \mathcal{C}_{nR_2}^n$ and we complete the proof. $\qquad\square$

Now we analyze the properties of $\bar{f}^{\text{nc-sc-sg}}$ and $\bar{f}_m^{\text{nc-sc-sg}}$.

**Lemma 11.** $\bar{f}^{\text{nc-sc-sg}}$ and $\bar{f}_m^{\text{nc-sc-sg}}$ satisfy the following.

    i. $\bar{f}_m^{\text{nc-sc-sg}}(\boldsymbol{0}, \boldsymbol{0}) - \inf_{\boldsymbol{x} \in \mathcal{C}_{R_1}^T, \boldsymbol{z} \in \mathcal{C}_{R_1}^{T-1}} \bar{f}_m^{\text{nc-sc-sg}}(\boldsymbol{x}, \boldsymbol{z}) \le 12T.$

    ii. $\bar{f}^{\text{nc-sc-sg}}$ is $\ell_0$-smooth for some numerical constant $\ell_0$.

    iii. $\bar{f}_m^{\text{nc-sc-sg}}$ is $\ell_m$-smooth for some numerical constant $\ell_m \ge 1$.

    iv. For all $\boldsymbol{x}, \boldsymbol{z}, \bar{\boldsymbol{y}}$, $\left\| \nabla \bar{f}^{\text{nc-sc-sg}}(\boldsymbol{x}, \boldsymbol{z}; \bar{\boldsymbol{y}}) \right\|_\infty \le G$ for some numerical constant $G$.

*Proof of Lemma 11.* Note that $\mathcal{C}_{R_1}^T \times \mathcal{C}_{R_1}^{T-1} \subset \mathbb{R}^T \times \mathbb{R}^{T-1}$. Then *i* and *ii* are direct corollaries of Lemma 9. We can prove *iii* in the same way as *ii*. It is also straightforward to verify *iv* given Lemma 6.*iii* and *iv* and noting the infinity norms of $\boldsymbol{x}$, $\boldsymbol{z}$, and $\bar{\boldsymbol{y}}$ are all bounded. $\qquad\square$

The lemma below shows we cannot find a good solution unless the end of the chain is reached.

**Lemma 12.** If $|z_i| < 1$ for some $i \le T$, then $(\boldsymbol{x}, \boldsymbol{z})$ is not a 1/3-stationary point of $\bar{f}_m^{\text{nc-sc-sg}}$.

*Proof of Lemma 12.* Let $1 < j \le i$ to be the smallest $j$ for which $|z_j| < 1$. Similar to the proof of Lemma 8, noting $\bar{f}_m^{\text{nc-sc}} = \bar{f}_m^{\text{nc-sc-sg}}$, we have

$$
\begin{aligned}
\frac{\partial \bar{f}_m^{\text{nc-sc-sg}}(\boldsymbol{x}, \boldsymbol{z})}{\partial x_{j-1}} &= p(x_{j-1}, z_{j-1}) + 12\left(x_{j-1} - \frac{1}{2}z_j\right), \\
\frac{\partial \bar{f}_m^{\text{nc-sc-sg}}(\boldsymbol{x}, \boldsymbol{z})}{\partial z_j} &= q(x_j, z_j) - 6\left(x_{j-1} - \frac{1}{2}z_j\right),
\end{aligned}
$$

where

$$-5 < p(x_{j-1}, z_{j-1}) < 0, \quad -20 < q(x_j, z_j) < 0.$$

There are two possible cases

1. If $|x_{j-1}| < 1$, we know $p(x_{j-1}, z_{j-1}) < -1$ by Lemma 6.*ii*. Then
$$\frac{\partial \bar{f}_m^{\text{nc-sc-sg}}(\boldsymbol{x}, \boldsymbol{z})}{\partial x_{j-1}} + 2 \cdot \frac{\partial \bar{f}_m^{\text{nc-sc-sg}}(\boldsymbol{x}, \boldsymbol{z})}{\partial z_j} = p(x_{j-1}, z_{j-1}) + 2q(x_j, z_j) < -1.$$
Therefore we can bound
$$\max\left\{ \left| \frac{\partial \bar{f}_m^{\text{nc-sc-sg}}(\boldsymbol{x}, \boldsymbol{z})}{\partial x_{j-1}} \right|, \left| \frac{\partial \bar{f}_m^{\text{nc-sc-sg}}(\boldsymbol{x}, \boldsymbol{z})}{\partial z_j} \right| \right\} > \frac{1}{3}.$$
Suppose $u$ is one of $x_{j-1}$ and $z_j$ such that $\left| \frac{\partial \bar{f}_m^{\text{nc-sc-sg}}(\boldsymbol{x}, \boldsymbol{z})}{\partial u} \right| > 1/3$. We also know $|u| < 1$. Let $\ell_m$ be the smoothness parameter of $\bar{f}_m^{\text{nc-sc-sg}}$ defined in Lemma 11.*iii*. Define
$$u' := u - \frac{1}{\ell_m} \frac{\partial \bar{f}_m^{\text{nc-sc-sg}}(\boldsymbol{x}, \boldsymbol{z})}{\partial u}. \tag{12}$$

i. If $|u'| \leq R_1$, we have
$$\ell_m \left| \mathsf{P}_{\mathcal{C}^1_{R_1}}(u') - u \right| = \ell_m |u' - u| = \left| \frac{\partial \bar{f}^{\text{nc-sc-sg}}_m(\boldsymbol{x}, \boldsymbol{z})}{\partial u} \right| > 1/3.$$

ii. If $|u'| > R_1$, we know that $\left| \mathsf{P}_{\mathcal{C}^1_{R_1}}(u') \right| = R_1$. Then we have
$$\ell_m \left| \mathsf{P}_{\mathcal{C}^1_{R_1}}(u') - u \right| > \ell_m(R_1 - 1) \geq 1.$$

2. If $x_{j-1} \geq 1$, we have $12(x_{j-1} - \frac{1}{2}z_j) > 6$. Since $-5 < p(x_{j-1}, z_{j-1}) < 0$, we must have
$$\frac{\partial \bar{f}^{\text{nc-sc}}_m(\boldsymbol{x}, \boldsymbol{z})}{\partial x_{j-1}} > 1.$$

Similar to case 1, we use $u$ to denote $x_{j-1}$ and define $u'$ as in (12). We know $u' < u$. Therefore

i. If $|u'| \leq R_1$, we have
$$\ell_m \left| \mathsf{P}_{\mathcal{C}^1_{R_1}}(u') - u \right| = \left| \frac{\partial \bar{f}^{\text{nc-sc-sg}}_m(\boldsymbol{x}, \boldsymbol{z})}{\partial u} \right| > 1.$$

ii. If $u' < -R_1$, we know that $\mathsf{P}_{\mathcal{C}^1_{R_1}}(u') = -R_1$. Then we have
$$\ell_m \left| \mathsf{P}_{\mathcal{C}^1_{R_1}}(u') - u \right| > \ell_m(R_1 + 1) \geq 1.$$

3. If $x_{j-1} \leq -1$, we have we have $12(x_{j-1} - \frac{1}{2}z_j) < -6$. Since $-5 < p(x_{j-1}, z_{j-1}) < 0$, we must have
$$\frac{\partial \bar{f}^{\text{nc-sc}}_m(\boldsymbol{x}, \boldsymbol{z})}{\partial x_{j-1}} < -6 < -1.$$

Then similar to case 2, we can show $\ell_m \left| \mathsf{P}_{\mathcal{C}^1_{R_1}}(u') - u \right| > \ell_m(R_1 + 1) \geq 1$.

To sum up, we have
$$\ell_m \left\| \mathsf{P}_{\mathcal{C}^T_{R_1} \times \mathcal{C}^{T-1}_{R_1}} \left( (\boldsymbol{x}, \boldsymbol{z}) - \frac{1}{\ell_m} \nabla \bar{f}^{\text{nc-sc-sg}}_m(\boldsymbol{x}, \boldsymbol{z}) \right) - (\boldsymbol{x}, \boldsymbol{z}) \right\|_2 \geq \ell_m \left| \mathsf{P}_{\mathcal{C}^1_{R_1}}(u') - u \right| > 1/3,$$

i.e., $(\boldsymbol{x}, \boldsymbol{z})$ is not a $1/3$-stationary point of $\bar{f}^{\text{nc-sc-sg}}_m$.

$\qquad \square$

With all the lemmas above, we are ready to prove Theorem 2.

*Proof of Theorem 2.* Similar to the proof of Theorem 1, we show the lower bound by appropriately rescaling $\bar{f}^{\text{nc-sc-sg}}$ as well as its domain. Formally, define $f^{\text{nc-sc-sg}} : \left( \mathcal{C}^T_{\lambda R_1} \times \mathcal{C}^{T-1}_{\lambda R_1} \right) \times \mathcal{C}^{n(T-1)}_{\lambda n R_2} \to \mathbb{R}$ as
$$f^{\text{nc-sc-sg}}(\boldsymbol{x}, \boldsymbol{z}; \bar{\boldsymbol{y}}) = \frac{L\lambda^2}{\ell_0} \bar{f}^{\text{nc-sc-sg}} \left( \frac{\boldsymbol{x}}{\lambda}, \frac{\boldsymbol{z}}{\lambda}; \frac{\bar{\boldsymbol{y}}}{\lambda} \right),$$

where $\lambda > 0$ is some parameter to be determined later and $\ell_0$ is the smoothness parameter defined in Lemma 11.*ii*. Note that we can show
$$f^{\text{nc-sc-sg}}_m(\boldsymbol{x}, \boldsymbol{z}) := \max_{\bar{\boldsymbol{y}} \in \mathcal{C}^{n(T-1)}_{\lambda n R_2}} f^{\text{nc-sc-sg}}(\boldsymbol{x}, \boldsymbol{z}; \bar{\boldsymbol{y}})$$
$$= \frac{L\lambda^2}{\ell_0} \max_{\boldsymbol{u} \in \mathcal{C}^{n(T-1)}_{n R_2}} \bar{f}^{\text{nc-sc-sg}} \left( \frac{\boldsymbol{x}}{\lambda}, \frac{\boldsymbol{z}}{\lambda}; \boldsymbol{u} \right)$$
$$= \frac{L\lambda^2}{\ell_0} \bar{f}^{\text{nc-sc-sg}}_m \left( \frac{\boldsymbol{x}}{\lambda}, \frac{\boldsymbol{z}}{\lambda} \right)$$

which means the order of maximization and rescaling can be interchanged. After the rescaling, $f_m^{\text{nc-sc-sg}}$ is still a zero-chain. Note that $f_m^{\text{nc-sc-sg}}$ is $\ell_m L/\ell_0$-smooth. When $z_T = 0$, by Lemma 12,

$$
\begin{aligned}
&\frac{\ell_m L}{\ell_0} \left\| \mathsf{P}_{\mathcal{C}_{\lambda R_1}^T \times \mathcal{C}_{\lambda R_1}^{T-1}} \left( (\boldsymbol{x}, \boldsymbol{z}) - \frac{\ell_0}{\ell_m L} \nabla f_m^{\text{nc-sc-sg}}(\boldsymbol{x}, \boldsymbol{z}) \right) - (\boldsymbol{x}, \boldsymbol{z}) \right\|_2 \\
=&\frac{\ell_m L}{\ell_0} \left\| \mathsf{P}_{\mathcal{C}_{\lambda R_1}^T \times \mathcal{C}_{\lambda R_1}^{T-1}} \left( \lambda \left( \frac{\boldsymbol{x}}{\lambda}, \frac{\boldsymbol{z}}{\lambda} \right) - \frac{\lambda}{\ell_m} \nabla \bar{f}_m^{\text{nc-sc-sg}} \left( \frac{\boldsymbol{x}}{\lambda}, \frac{\boldsymbol{z}}{\lambda} \right) \right) - \lambda \left( \frac{\boldsymbol{x}}{\lambda}, \frac{\boldsymbol{z}}{\lambda} \right) \right\|_2 \\
=&\frac{L\lambda}{\ell_0} \ell_m \left\| \mathsf{P}_{\mathcal{C}_{R_1}^T \times \mathcal{C}_{R_1}^{T-1}} \left( \left( \frac{\boldsymbol{x}}{\lambda}, \frac{\boldsymbol{z}}{\lambda} \right) - \frac{1}{\ell_m} \nabla \bar{f}_m^{\text{nc-sc-sg}} \left( \frac{\boldsymbol{x}}{\lambda}, \frac{\boldsymbol{z}}{\lambda} \right) \right) - \left( \frac{\boldsymbol{x}}{\lambda}, \frac{\boldsymbol{z}}{\lambda} \right) \right\|_2 \\
>&\frac{L\lambda}{3\ell_0}.
\end{aligned}
$$

Choosing $\lambda = \frac{6\ell_0 \epsilon}{L}$ guarantees such $(\boldsymbol{x}, \boldsymbol{z})$ is not a $2\epsilon$-stationary point of $f^{\text{nc-sc-sg}}$.

Now we check $f^{\text{nc-sc-sg}} \in \mathcal{F}(L, \mu, \Delta)$. Note that

$$
\nabla^2 f^{\text{nc-sc-sg}}(\boldsymbol{x}, \boldsymbol{z}; \bar{\boldsymbol{y}}) = \frac{L}{\ell_0} \nabla^2 \bar{f}^{\text{nc-sc-sg}} \left( \frac{\boldsymbol{x}}{\lambda}, \frac{\boldsymbol{z}}{\lambda}; \frac{\bar{\boldsymbol{y}}}{\lambda} \right).
$$

We know the smoothness parameter of $f^{\text{nc-sc-sg}}$ is $L$ and the strong concavity parameter is $\frac{L}{\ell_0 n^3}$. Therefore we should choose

$$
n = \left\lfloor \left( \frac{L}{\mu \ell_0} \right)^{1/3} \right\rfloor
$$

to make $f^{\text{nc-sc-sg}}$ $\mu$-strongly concave in $\bar{\boldsymbol{y}}$. Then it suffices to show $f_m^{\text{nc-sc-sg}}(\boldsymbol{0}, \boldsymbol{0}) - \inf_{\boldsymbol{x}, \boldsymbol{z}} f_m^{\text{nc-sc-sg}}(\boldsymbol{x}, \boldsymbol{z}) \leq \Delta$. By Lemma 9,

$$
f_m^{\text{nc-sc-sg}}(\boldsymbol{0}, \boldsymbol{0}) - \inf_{\boldsymbol{x}, \boldsymbol{z}} f_m^{\text{nc-sc-sg}}(\boldsymbol{x}, \boldsymbol{z}) = \frac{L\lambda^2}{\ell_0} \left( \bar{f}_m^{\text{nc-sc-sg}}(\boldsymbol{0}, \boldsymbol{0}) - \inf_{\boldsymbol{x}, \boldsymbol{z}} \bar{f}_m^{\text{nc-sc-sg}} \left( \frac{\boldsymbol{x}}{\lambda}, \frac{\boldsymbol{z}}{\lambda} \right) \right) \leq \frac{12LT\lambda^2}{\ell_0},
$$

which is no greater than $\Delta$ if choosing

$$
T = \left\lfloor \frac{\ell_0 \Delta}{12L\lambda^2} \right\rfloor = \left\lfloor \frac{L\Delta}{432\ell_0 \epsilon^2} \right\rfloor.
$$

Now we construct the stochastic gradient oracle in the same way as [3]. We perturb the gradient only on the next coordinate to discover, so that we reveal its value with probability $p$. Let $i^*(\boldsymbol{x}, \boldsymbol{z}; \bar{\boldsymbol{y}})$ denote the next coordinate to discover in the zero-chain in Figure 1(c). Precisely, we set the stochastic gradient to be

$$
\boldsymbol{g}(\boldsymbol{x}, \boldsymbol{z}; \bar{\boldsymbol{y}}; \xi)_i = \begin{cases} \frac{\xi}{p} \nabla_i f^{\text{nc-sc-sg}}(\boldsymbol{x}, \boldsymbol{z}; \bar{\boldsymbol{y}}) & \text{if } i = i^*(\boldsymbol{x}, \boldsymbol{z}; \bar{\boldsymbol{y}}) \\ \nabla_i f^{\text{nc-sc-sg}}(\boldsymbol{x}, \boldsymbol{z}; \bar{\boldsymbol{y}}) & \text{otherwise,} \end{cases}
$$

where $\xi \sim \text{Bernoulli}(p)$. By Lemma 5, $f^{\text{nc-sc-sg}}$ is a probability-$p$ zero-chain with this oracle which has variance bounded by

$$
\mathbb{E} \left[ \| \boldsymbol{g}(\boldsymbol{x}, \boldsymbol{z}; \bar{\boldsymbol{y}}; \xi) - \nabla f^{\text{nc-sc-sg}}(\boldsymbol{x}, \boldsymbol{z}; \bar{\boldsymbol{y}}) \|_2^2 \right] \leq \left( \frac{GL\lambda}{\ell_0} \right)^2 \frac{1-p}{p} = 36\epsilon^2 G^2 \frac{1-p}{p}.
$$

Hence, the variance is no greater than $\sigma^2$ if $p = \min\{1, \frac{36\epsilon^2 G^2}{\sigma^2}\}$. By Lemma 3, with probability $1 - \delta$, $z_T^t = 0$ for all

$$
t \leq \frac{n(T-1) - \log(1/\delta)}{2p}.
$$

Then taking $\delta = 1/2$ yields that whenever

$$
t \leq \frac{n(T-1) - 1}{2 \frac{36\epsilon^2 G^2}{\sigma^2}} = \frac{c' n T \sigma^2}{\epsilon^2 G^2} = \frac{c_0' L \Delta \sigma^2 \kappa^{1/3}}{\epsilon^4},
$$

for some constant $c', c_0 > 0$, we have

$$\mathbb{E}\left[\frac{\ell_m L}{\ell_0}\left\|\mathsf{P}_{\mathcal{C}_{\lambda R_1}^T \times \mathcal{C}_{\lambda R_1}^{T-1}}\left((\boldsymbol{x}, \boldsymbol{z}) - \frac{\ell_0}{\ell_m L}\nabla f_m^{\text{nc-sc-sg}}(\boldsymbol{x}, \boldsymbol{z})\right) - (\boldsymbol{x}, \boldsymbol{z})\right\|_2\right] \geq \frac{1}{2} \cdot 2\epsilon = \epsilon.$$

That is, $(\boldsymbol{x}^t, \boldsymbol{z}^t)$ is not an $\epsilon$-stationary point. So far we have derived a lower bound of $\Omega(\frac{L\Delta\sigma^2\kappa^{1/3}}{\epsilon^4})$.
Note that the deterministic lower bound is $\Omega(\frac{L\Delta\sqrt{\kappa}}{\epsilon^2})$ which is a special case of the stochastic setting.
Therefore we derive a lower bound of

$$\Omega\left(L\Delta\max\left\{\frac{\sqrt{\kappa}}{\epsilon^2}, \frac{\kappa^{1/3}\sigma^2}{\epsilon^4}\right\}\right) = \Omega\left(L\Delta\left(\frac{\sqrt{\kappa}}{\epsilon^2} + \frac{\kappa^{1/3}\sigma^2}{\epsilon^4}\right)\right).$$

$\square$