# OpenReview forum: "Complexity Lower Bounds for Nonconvex-Strongly-Concave Min-Max Optimization"
_NeurIPS.cc/2021/Conference — NeurIPS 2021 Poster_

### Official Review · Reviewer_aNZZ · 2021-07-10

**Rating:** 7
**Confidence:** 4

**Summary:**

This paper proposed a novel first-order oracle lower complexity bound for finding stationary points of smooth min-max optimization problems where the objective function is nonconvex in min variable, and strongly concave in max variable. The lower bound of $\Omega(\sqrt{\kappa}\epsilon^{-2})$ in deterministic setting matches the recent upper bound (up to log factors), hence is nearly optimal. Another lower bound of $\Omega(\sqrt{\kappa}\epsilon^{-2} + \kappa^{1/3}\epsilon^{-4})$ is provided for stochastic oracles, suggesting that the best-known upper bound of $O(\kappa^3\epsilon^{-4})$ is far from optimal in the condition number dependence. Overall, this paper is nicely written. The authors did a fantastic job in reviewing SOTA results, and put the newly derived lower complexity bounds in context. High-level explanations of the construction idea was provided, which makes it much easier to understand the proof and the complexity bound. The obtained results, to the best of my knowledge, are new and significant.

**Limitations And Societal Impact:**

This is a great paper which presents its ideas clearly and proves a good result, and it should be accepted to NeurIPS. I have no further complaints.

**Main Review:**

I only have some minor comments.

Throughout the paper, the authors assumed the algorithm to be zero-respecting (Definition 4) and claim that there is a standard reduction from a lower bound for zero-respecting algorithms to that for arbitrary deterministic algorithms [3][5]. However, these cited papers are mostly based on the classical resisting oracles initialized by Nemirovski, A.S (to my knowledge)

Nemirovski, A.S.: Information-based complexity of linear operator equations. J. Complexity 8(2), 153–175 (1992)

Nemirovsky, A.: On optimality of krylov’s information when solving linear operator equations. J. Complexity 7(2), 121–130 (1991)

Such resisting oracle seems not easily extended to study general min-max optimization problems beyond the quadratic case; see the following article.

Ouyang, Y. and Xu, Y. Lower complexity bounds of first-order methods for convex-concave bilinear saddle-point problems, Mathematical Programming volume 185,1–35 (2021).

It would be better to elaborate this issue more clearly in the paper.

Theorem 2 is particularly interesting to me, especially the condition number dependence of $\kappa^{1/3}$. Any intuition behind the order? Any insights on which algorithm might achieve this lower bound?



**Time Spent Reviewing:**

4 hours

---

> ### Author Response · Authors · 2021-08-09
> **Response to Reviewer aNZZ**
>
> Thank you for your detailed comments!
>
> For the first question, please check Proposition 1 as well as its proof in the following paper [Carmon et al. 2019] for detailed analysis. It reduces the complexity lower bound for zero-respecting algorithms to that ofarbitrary deterministic algorithms. Although this proposition is for minimization problems, we believe it should not be very hard to extend such analysis to min-max optimization problems and will leave it as future work.
>
> Yair Carmon, John C Duchi, Oliver Hinder, and Aaron Sidford. Lower bounds for finding stationary points i. Mathematical Programming, pages 1–50, 2019.
>
> Regarding the $\kappa^{1/3}$ rate in Theorem 2, it is derived by our technical construction and we do not know whether it is tight. Closing this gap should be an interesting and important open problem that is beyond the scope of this paper.

---

### Official Review · Reviewer_wgu3 · 2021-07-15

**Rating:** 7
**Confidence:** 4

**Summary:**

This paper consider the lower oracle complexity bound of NC-SC min-max problems in deterministic and stochastic settings. Based on the zero-chain argument in previous first-order optimization literature, the authors proposed a novel hard instance construction and shows that the lower bounds are $\mathcal{O}(\sqrt{\kappa}\epsilon^{-2})$ for deterministic setting and $\mathcal{O}(\sqrt{\kappa}\epsilon^{-2}+\kappa^{1/3}\sigma^2\epsilon^{-4})$ in stochastic case.

**Limitations And Societal Impact:**

This work does not involve issues related to the negative impact of society.

**Main Review:**

As pointed out by authors, exploring the lower bound of NC-SC min-max optimization problem is natural and imporatant following many recent works on nonconvex optimization and NC-SC algorithms works. The flow of the paper is very clear.

The main technique of the lower bound proof comes from previous lower bound works on convex optimization [34,36] and nonconvex optimization [3,5,6,13], i.e. the zero-chain argument. While the detailed hard instance construction, in my opinion, is novel and inspiring, especially the design of the subchain, which formulate a multiplication in the dimension numbers to get the final lower complexity results.

This paper considers deterministic and stochastic cases, while as authors pointed out, there is already concurrent works on deterministic case with the same result, the analysis in the stochastic case should be the first nontrivial result specifically in NC-SC min-max literature, even though there is still a large gap in terms of the dependence on the condition number ($\kappa^{1/3}$ vs. $\kappa^3$), this result should still be important.

Some problems:
1. Line 599 in the Appendix, is it $||u'||$ and $||P_C(u')||$, rather than solely $u'$ and $P_C(u')$? Also I think it should be $||\cdot||$, rather than the absolute value sign $|\cdot|$.

**Time Spent Reviewing:**

8 (sum of finite intervals)

---

> ### Author Response · Authors · 2021-08-09
> **Response to Reviewer wgu3**
>
> Thank you so much for your detailed comments! Let us answer your problem.
>
> Our proof around Line 599 should be correct. First, please note $u’$ is actually a scalar. Because $u’$ is defined in Equation (12) in Line 592, where $u:=x_{j-1}$ in case 2 (see Line 596) which is the $j-1$-th coordinate of ${x}$, and thus, a scalar. Second, in case 2, we do not need to discuss the case where $u’>R_1$ because we always have $u’<u\le R_1$ (see Equation (12) and the inequality right above Line 596). However, we do have a typo in the inequalities right above Line 596 and Line 602: it should be $\bar{f}_m^{\text{nc-sc-sg}}$ rather than $\bar{f}_m^{\text{nc-sc}}$. We will fix it in the revision.

---

> > ### Comment · Reviewer_wgu3 · 2021-08-19
> > **Feedback**
> >
> > Thank you for the clarification, I will keep my score.

---

### Official Review · Reviewer_SL4u · 2021-07-18

**Rating:** 6
**Confidence:** 3

**Summary:**

The paper aims to provide lower complexity bounds for nonconvex-strongly-concave (NC-SC) minimax optimization problems. It shows $\Omega(\sqrt{\kappa}\epsilon^{-2}) for the deterministic setting and $\Omega\left(\sqrt{\kappa} \epsilon^{-2}+\kappa^{1 / 3} \epsilon^{-4}\right)$ for the stochastic setting. The lower bound for the deterministic setting has been nearly matched by several upper bounds, so it should be tight.

**Limitations And Societal Impact:**

Yes.

**Main Review:**

$\textbf{Originality}$: I believe the work is original. The idea in section 5.1 to glue two zero chains is novel and interesting to me.

$\textbf{Clarity}$: The structure of the paper is well-organized. The writing is clear, except a few grammar errors, such as in line 177-178.

$\textbf{Significance}$: To discover the lower complexity bound in NC-SC is an important task. Stochastic algorithms are widely used in practice, but the lower bound in this setting is under-explored. This paper is an initial step.

$\textbf{Questions and comments}$:

1. One of my concerns is the contribution of the results. Except [Zhang et. al., 2021], the following paper appearing on arxiv this April also provides the same results for the deterministic NC-SC problems:
 "Yuze Han, Guangzeng Xie, and Zhihua Zhang. Lower complexity bounds of finite-sum optimization problems:
The results and construction. arXiv preprint arXiv:2103.08280, 2021."
Therefore, the lower bound for deterministic setting can be showed by various methods. For the result of stochastic setting, although the paper shows the first result with the dependency in $\kappa$, it does not seem tight to me, and the gap with the upper bound remains. So I am not sure about the potential of this method.

2. The stationarity measure used in the paper is $L_m\left\|P_{\mathcal{X}}[x-(1 / L_m) \nabla f(x)]-x \right\|_{2} \leq \epsilon$, where $L_m$ is the smoothness constant of the function $f_m = \max_x f(x,y)$. I am not sure whether there is existing upper bound that can nearly achieve this when $x$ is constrained. On the other hand, the upper bound it compares to requires the domain of $y$ to be bounded, while the hard instance the paper uses in the deterministic lower bound has unconstrained $y$.

3. Is it possible to extend the analysis to stochastic setting with the averaged smoothness assumption?

**Time Spent Reviewing:**

3

---

> ### Author Response · Authors · 2021-08-09
> **Response to Reviewer SL4u**
>
> Thank you so much for your detailed comments.
>
> 1.**We were not aware of the work [Han et al] and will discuss that paper in our revision. Our work is done concurrently with these two and published a few weeks later.** We will clarify this via explaining the difference between our proof and the proof by the other two concurrent works. In fact, precisely because of the difference in proof techniques, we can construct a lower bound for the stochastic setting .
>
> First note that **the constructions of [Zhang et al. ‘21] and [Han et al. ‘21] are almost the same** for the general nonconvex, strongly concave (NC-SC) optimization problem. But **ours is different** from theirs. Below,we clarify the difference.
>
> Zhang/Han et al’s construction, $f(x,y)$, builds on the “hard instance” for nonconvex minimization problems in [Carmon et al. ‘19b]. Essentially,  they choose $f_m(x):=\max_{y}f(x,y)$ to be the hard instance in [Carmon et al‘19b]. When designing such $f$, they are inspired by the hard instance in [Ouyang and Xu 2019] for convex-strongly-concave min-max problems. Their example can also be viewed as a simple **addition** of the quadratic hard instance for convex-strongly-concave problems and a coordinate-wise nonconvex function. Informally speaking, their $\sqrt{\kappa}$ factor comes from the rate in [Ouyang and Xu 2019].
>
> In contrast, our idea is inspired by **another** Carmon et al’s work, [Carmon et al. ‘19a] (not [Carmon et al.’19b]), and not directly related to [Ouyang and Xu 2019]. We use the example in [Carmon et al. ‘19a] as the main body of our zero-chain and  insert many “strongly concave” sub-chains to **enlarge the length of the chain to get the additional $\sqrt{\kappa}$ factor.** Our example can be viewed as a **structural composition** of the nonconvex and quadratic components.  In our view, our construction has the following advantage:
> In our example, **$x$ and $y$ are coupled in a much weaker fashion**: only a small fraction of $y$’s coordinates are directly connected to $x$, whereas in their construction, all the coordinates of $y$ are connected to $x$. **This provides more freedom to modify the components that depend only on $y$ when we want to extend the result to other settings. This allows us, as an example,  to modify the expression of the sub-chains to obtain our lower bound in the stochastic setting.**
>
> 2.For the first question, the answer is yes. [Lin et al. 2020b]’s upper bounds apply to the setting where $x$ is constrained. In fact, they use the same optimality measure. This measure is not limited to constrained settings, it reduces to $\|\nabla f\|\le \epsilon$ for unconstrained problems.
>
> For the second question, our analysis also holds if constraining $y$ within a large enough hypercube, as long as for every $x\in\mathcal{X}$, $y^*(x)=\text{argmax}_y f(x,y)$ is inside the constraint. The analysis for this constrained setting is similar to what we did in the stochastic setting.
>
> 3.This is a great question. In our standard stochastic case (rather than the stricter case as in the SPIDER paper), there is no component function, and hence the smoothness assumption is on the expected objective. In the case when we assume that the stochastic function is composed of individual smooth functions, our lower bound is probably not tight on epsilon dependence.  We are not sure about the answer but it is an interesting and important open problem.
>
> Tianyi Lin, Chi Jin, and Michael I Jordan. Near-optimal algorithms for minimax optimization. In Conference on Learning Theory, pages 2738–2779. PMLR, 2020b.
>
> Yair Carmon, John C Duchi, Oliver Hinder, and Aaron Sidford. Lower bounds for finding stationary points **i**. Mathematical Programming, pages 1–50, **2019a**.
>
> Yair Carmon, John C Duchi, Oliver Hinder, and Aaron Sidford. Lower bounds for finding stationary points **ii**: first-order methods. Mathematical Programming, pages 1–41, **2019b**.
>
> Yuyuan Ouyang and Yangyang Xu. Lower complexity bounds of first-order methods for convex-concave bilinear saddle-point problems. Mathematical Programming, pages 1–35, 2019.

---

> > ### Comment · Reviewer_SL4u · 2021-08-19
> > **stationarity measure**
> >
> > 1. I would like to increase my rate given the interesting idea of sub-chain construction, although the first part of the result has already appeared before.
> >
> > 2. It seems to me the stationarity measures in this paper and [Lin et al., 2020b] are not exactly the same. Here in $L_{m}\left\Vert P_{X}\left[x-\left(1 / L_{m}\right) \nabla f(x)\right]-x\right\Vert _{2}$, $L_m$ is the smoothness of the primal function, while [Lin et al., 2020b] uses the smoothness of $f$. This is just a minor comment.

---

> > > ### Author Response · Authors · 2021-08-19
> > > **Thanks!**
> > >
> > > Thank you for increasing the score!
> > >
> > > [Lin et al., 2020b] actually considers two kinds of stationary measures: the stationarity of $f$ and the stationarity of the primal function $f_m$.  The complexity bounds for these two measures are the same in the nonconvex-strongly-concave setting (see Table 2 in their paper). Our definition is the same as the second one which uses the smoothness of $f_m$. Also, in our example, these two smoothness parameters are actually nearly equal up to a constant factor, see Lemma 11.$ii$ and 11.$iii$ around Line 577. So our analysis applies to both smoothness parameters.

---

### Official Review · Reviewer_KB1h · 2021-07-26

**Rating:** 6
**Confidence:** 4

**Summary:**

This paper establishes lower bounds for finding approximate stationary points of min_x max_y f(x;y), where f is nonconvex in x and strongly concave in y. With deterministic gradients, the lower bound matches the convergence rate of the accelerated "Minimax-PPA" algorithm from [Lin et al. '20b], settling the oracle complexity in this setting, combining the bump-function construction of [Carmon et al. '19] with the classic zero-chain construction for convex optimization. A stochastic gradient oracle is constructed, following the same stochastic zero-chain construction as [Arjevani et al. '19], providing a lower bound scaling as \kappa^{1/3} / \eps^4 times the variance.

**Limitations And Societal Impact:**

The work is purely theoretical in nature. The discussion of limitations is adequate.

**Main Review:**

- This paper settles the oracle complexity of a problem that has seen a flurry of recent theoretical and empirical interest. It fills a clear gap in the literature, and leaves an interesting question about the possible suboptimality of the best known algorithm for the stochastic setting. The exposition is very clear.
- My overall assessment of this paper depends on whether or not the cited work [Zhang et al. 21] ("The complexity of nonconvex-strongly-concave minimax optimization"), which appeared on arXiv on March 29, 2021, counts as concurrent. It is claimed that the proofs and construction are different, but from what I could tell, Zhang et al. also arrive at a "zero-chain within zero-chain" construction, and the proof is not that conceptually different.
- [Zhang et al. '21] themselves refer to concurrent work which was posted 2 weeks prior ([Han et al. '21], "Lower Complexity Bounds of Finite-Sum Optimization Problems: The Results and Construction"), which also has overlapping material.
- The treatment of the pure stochastic (as opposed to finite-sum) case is still new, and the modification of the construction to get a stochastic oracle with bounded variance is fairly subtle. However, in my opinion this does not suffice as a standalone novelty.
- If the authors could articulate the differences vs. these works more clearly, this would be helpful towards reaching a conclusion independent of the concurrency issue.

Small typos:
- 247: "effictive"
- 248: "suppose" -> "supposing"

**Time Spent Reviewing:**

5

---

> ### Author Response · Authors · 2021-08-09
> **Response to Reviewer KB1h**
>
> Thank you so much for your detailed comments. **We were not aware of the work [Han et al] and will discuss that paper in our revision. Our work is done concurrently with these two and published a few weeks later.** We will clarify this via explaining the difference between our proof and the proof by the other two concurrent works. In fact, precisely because of the difference in proof techniques, we can construct a lower bound for the stochastic setting .
>
> First note that **the constructions of [Zhang et al. ‘21] and [Han et al. ‘21] are almost the same** for the general nonconvex, strongly concave (NC-SC) optimization problem. But **ours is different** from theirs. Below,we clarify the difference.
>
> Zhang/Han et al’s construction, $f(x,y)$, builds on the “hard instance” for nonconvex minimization problems in [Carmon et al. ‘19b]. Essentially,  they choose $f_m(x):=\max_{y}f(x,y)$ to be the hard instance in [Carmon et al‘19b]. When designing such $f$, they are inspired by the hard instance in [Ouyang and Xu 2019] for convex-strongly-concave min-max problems. Their example can also be viewed as a simple **addition** of the quadratic hard instance for convex-strongly-concave problems and a coordinate-wise nonconvex function. Informally speaking, their $\sqrt{\kappa}$ factor comes from the rate in [Ouyang and Xu 2019].
>
> In contrast, our idea is inspired by **another** Carmon et al’s work, [Carmon et al. ‘19a] (not [Carmon et al.’19b]), and not directly related to [Ouyang and Xu 2019]. We use the example in [Carmon et al. ‘19a] as the main body of our zero-chain and  insert many “strongly concave” sub-chains to **enlarge the length of the chain to get the additional $\sqrt{\kappa}$ factor.** Our example can be viewed as a **structural composition** of the nonconvex and quadratic components.  In our view, our construction has the following advantage:
>
> In our example, **$x$ and $y$ are coupled in a much weaker fashion**: only a small fraction of $y$’s coordinates are directly connected to $x$, whereas in their construction, all the coordinates of $y$ are connected to $x$. **This provides more freedom to modify the components that depend only on $y$ when we want to extend the result to other settings. This allows us, as an example,  to modify the expression of the sub-chains to obtain our lower bound in the stochastic setting.**
>
> Finally,we thank the reviewer for finding the typos! We will fix  them in revision.
>
>
> Yair Carmon, John C Duchi, Oliver Hinder, and Aaron Sidford. Lower bounds for finding stationary points **i**. Mathematical Programming, pages 1–50, **2019a**.
>
> Yair Carmon, John C Duchi, Oliver Hinder, and Aaron Sidford. Lower bounds for finding stationary points **ii**: first-order methods. Mathematical Programming, pages 1–41, **2019b**.
>
> Yuyuan Ouyang and Yangyang Xu. Lower complexity bounds of first-order methods for convex-concave bilinear saddle-point problems. Mathematical Programming, pages 1–35, 2019.

---

> > ### Comment · Reviewer_KB1h · 2021-08-31
> > **Update**
> >
> > Thanks for the clarifications about the distinctions. After considering the response and the other reviews, I'm convinced that this construction is a nice technical improvement on top of [Zhang et al. ‘21] and [Han et al. ‘21]. If those two works are to be counted as prior work (given the gap between the time those were posted vs. the submission deadline for this venue), then this would decrease the novelty factor for this paper somewhat. However, the work is solid, and the improvement is nicely articulated. I have increased my score.

---

### Decision · Program_Chairs · 2021-09-27

**Decision:**

Accept (Poster)

**Comment:**

The paper provides a complexity lower bound for first-order optima for smooth non-convex and strongly concave min-max problems. The reviewers were unanimous in their appreciation of the construction of the lower bound and the result of the paper. The main concern with the paper is the presence of concurrent work which decreases the novelty of the result of the paper. The reviewers have nevertheless found the paper solid and the improvements/differences somewhat nicely articulated. I strongly suggest the authors to highlight the differences between the paper and the exisiting works clearly in the final version.